# ISR: Invertible Symbolic Regression

## Abstract

We introduce an Invertible Symbolic Regression (ISR) method, a machine learning technique that generates analytical relationships between inputs and outputs of a given dataset via invertible maps (or architectures). The proposed ISR method naturally combines the principles of Invertible Neural Networks (INNs) and Equation Learner (EQL), a neural network-based symbolic architecture for function learning. In particular, we transform the affine coupling blocks of INNs into a symbolic framework, resulting in an end-to-end differentiable symbolic invertible architecture that allows for efficient gradient-based learning. The proposed ISR framework also relies on sparsity promoting regularization, allowing the discovery of concise and interpretable invertible expressions. We show that ISR can serve as a (symbolic) normalizing flow for density estimation tasks. Furthermore, we highlight its practical applicability in solving inverse problems, including a benchmark inverse kinematics problem, and notably, a geoacoustic inversion problem in oceanography aimed at inferring posterior distributions of underlying seabed parameters from acoustic signals.

## 1 Introduction

In many applications in engineering and science, experts have developed theories about how measurable quantities result from system parameters, known as forward modeling. In contrast, the *Inverse Problem* aims to find unknown parameters of a system that lead to desirable observable quantities. A typical challenge is that numerous configurations of these parameters yield the same observable quantity, especially with underlying complicated nonlinear governing equations and where hidden parameters outnumber the observable variables. To tackle challenging and ill-posed inverse problems, a common method involves estimating a posterior distribution on the unknown parameters, given the observations. Such a probabilistic approach facilitates the uncertainty quantification by analyzing the diversity of potential inverse solutions.

An established computationally expensive approach in finding the posterior distribution is to directly generate samples using acceptance/rejection. In this scope, the Markov Chain Monte Carlo (MCMC) methods (Brooks et al., 2011; Andrieu et al., 2003; Doucet & Wang, 2005; Murphy, 2012; Goodfellow et al., 2016; Korattikara et al., 2014; Atchadé & Rosenthal, 2005; Kungurtsev et al., 2023) offer a strong alternative for achieving near-optimum Bayesian estimation (Constantine et al., 2016; Conrad et al., 2016; Dosso & Dettmer, 2011). However, MCMC methods can be inefficient MacKay (2003) as the number of unknown parameter increases.

When the likelihood is unknown or intractable, the Approximate Bayesian Computation (ABC) Csilléry et al. (2010) is often used to estimate the posterior distribution. However, similar to MCMC, this method also suffers from poor scalability Cranmer et al. (2020a); Papamakarios et al. (2019). A more efficient alternative is to approximate the posterior using a tractable distribution, i.e. the variational method Blei et al. (2017); Salimans et al. (2015); Wu et al. (2018). However, the performance of the variational method deteriorates as the true posterior becomes more complicated.

Neural networks have become popular for solving inverse problems due to their ability to effectively handle complex relationships. They can be used not only for finding point estimates but also in the Bayesian framework to estimate the posterior distribution. For instance, in the non-Bayesian setting, by leveraging the rich mathematical and physical theories behind the inverse problems, the works in (Ying, 2022; Fan & Ying, 2019; Khoo & Ying, 2019) developed novel neural network architectures for solving these problems

while reducing the reliance on large amounts of data. Moreover, in the Bayesian setting, the learning-enabled method for sampling distribution has gained attention as they shown to outperform traditional methods. One approach is to utilize learned surrogate models within traditional sampling methods such as MCMC. Since direct sampling of the posterior distribution requires many runs of the forward map, often a trained and efficient surrogate model is used instead of the exact model (Li et al., 2023). Surrogate models are considered as an efficient representation of the forward map trained on the data. Popular approaches include recently introduced Physics-Informed Neural Networks (Raissi et al., 2019). Also, invertible architectures are shown to be well-suited for solving inverse problems. Unlike classical Bayesian neural networks (Kendall & Gal, 2017), which directly tackle the ambiguous inverse problem, invertible Neural Networks (INNs) learn the forward process and utilize latent variables to capture otherwise lost information. The invertible structure of INNs implicitly learns the inverse process, providing the full posterior over parameter space and circumventing the challenge of defining a supervised loss in the inverse direction for direct posterior learning (Ardizzone et al., 2019a; Zhang & Curtis, 2021; Luce et al., 2023; Putzky & Welling, 2019; Guan et al.).

Due to the black-box nature of neural networks, it is beneficial to express the forward map symbolically instead for several reasons. First, Symbolic Regression (SR) can provide model interpretability, while understanding the inner workings of deep Neural Networks is challenging (Kim et al., 2020; Gilpin et al., 2018). Second, studying the symbolic outcome can lead to valuable insights and provide nontrivial relations and/or physical laws (Udrescu & Tegmark, 2020; Udrescu et al., 2020; Liu & Tegmark, 2021; Keren et al., 2023). Third, they may achieve better results than Neural Networks in out-of-distribution generalization Cranmer et al. (2020b). Fourth, unlike conventional regression methods, such as least squares (Wild & Seber, 1989), likelihood-based (Edwards, 1984; Pawitan, 2001; Tohme et al., 2023b), and Bayesian regression techniques (Lee, 1997; Leonard & Hsu, 2001; Vanslette et al., 2020; Tohme et al., 2020; Tohme, 2020), SR does not rely on a fixed parametric model structure.

The attractive properties of SR, such as interpretability, often come at high computational cost compared to standard Neural Networks. This is because SR optimizes for model structure and parameters simultaneously. Therefore, SR is thought to be NP-hard (Petersen et al., 2021; Virgolin & Pissis, 2022). However tractable solutions exist, that can approximate the optimal solution suitable for applications. For instance, genetic algorithms (Koza & Koza, 1992; Schmidt & Lipson, 2009; Tohme et al., 2023a; Orzechowski et al., 2018; La Cava et al., 2021) and machine learning algorithms, such as neural networks, and transformers (Sahoo et al., 2018; Jin et al., 2019; Udrescu et al., 2020; Cranmer et al., 2020b; Kommenda et al., 2020; Burlacu et al., 2020; Biggio et al., 2021; Mundhenk et al., 2021; Petersen et al., 2021; Valipour et al., 2021; Zhang et al., 2022; Kamienny et al., 2022), are used to solve SR efficiently.

**Related Works.** In recent years, a branch of Machine Learning methods has emerged and is dedicated to finding data-driven invertible maps. While they are ideal for data generation and inverse problem, they lack interpretablity. On the other hand, several methods have been developed to achieve interpretablity in representing the forward map via Symbolic Regression. Hence, it is natural to incorporate SR in the invertible map for the inverse problem. Next, we review the related works in the scope of this work.

*Normalizing Flows:* The idea of this class of methods is to train an invertible map such that in the forward problem, the input samples are mapped to a known distribution function, e.g. the normal distribution function. Then, the unknown distribution function is found by inverting the trained map with the normal distribution as the input. This procedure is called the normalizing flow technique (Rezende & Mohamed, 2015; Dinh et al., 2016; Kingma & Dhariwal, 2018; Durkan et al., 2019; Tzen & Raginsky, 2019; Kobyzev et al., 2020; Wang & Marzouk, 2022). This method has been used for re-sampling unknown distributions, e.g. Boltzmann generators (Noé et al., 2019), as well as density recovery such as AI-Feynmann (Udrescu & Tegmark, 2020; Udrescu et al., 2020).

*Invertible Neural Networks (INNs):* This method can be categorized in the class of normalizing flows. The invertibility of INN is rooted in their architecture. The most popular design is constructed by concatenating affine coupling blocks Kingma & Dhariwal (2018); Dinh et al. (2016; 2014), which limits the architecture of the neural network. INNs have been shown to be effective in estimating the posterior of the probabilistic inverse problems while outperforming MCMC, ABC, and variational methods. Applica-

tions include epidemiology Radev et al. (2021), astrophysics Ardizzone et al. (2019a), medicine Ardizzone et al. (2019a), optics Luce et al. (2023), geophysics Zhang & Curtis (2021); Wu et al. (2023), and reservoir engineering Padmanabha & Zabaras (2021). Compared to classical Bayesian neural networks for solving inverse problems, INNs lead to more accurate and reliable solutions as they leverage the better-understood forward process, avoiding the challenge of defining a supervised loss for direct posterior learning. Ardizzone et al. (2019a). However, similar to standard neural networks, INNs lead to a model that cannot be evaluated with interpretable mathematical formula.

*Equation Learner (EQL):* Among SR methods, the EQL network is one of the attractive methods since it incorporates gradient descent in the symbolic regression task for better efficiency (Martius & Lampert, 2016; Sahoo et al., 2018; Kim et al., 2020). EQL devises a neural network-based architecture for SR task by replacing commonly used activation functions with a dictionary of operators and use back-propagation for training. However, in order to obtain a symbolic estimate for the inverse problem efficiently, it is necessary to merge such efficient SR method with an invertible architecture, which is the goal of this paper.

**Our Contributions.** We present Invertible Symbolic Regression (ISR), a machine learning technique that identifies mathematical relationships that best describe the forward and inverse map of a given dataset through the use of invertible maps. ISR is based on an invertible symbolic architecture that bridges the concepts of Invertible Neural Networks (INNs) and Equation Learner (EQL), i.e. a neural network-based symbolic architecture for function learning. In particular, we transform the affine coupling blocks of INNs into a symbolic framework, resulting in an end-to-end differentiable symbolic inverse architecture. This allows for efficient gradient-based learning. The symbolic invertible architecture is easily invertible with a tractable Jacobian, which enables explicit computation of posterior probabilities. The proposed ISR method, equipped with sparsity promoting regularization, captures complex functional relationships with concise and interpretable invertible expressions. In addition, as a byproduct, we naturally extend ISR into a conditional ISR (cISR) architecture by integrating the EQL network within conditional INN (cINN) architectures present in the literature. We further demonstrate that ISR can also serve as a symbolic normalizing flow (for density estimation) in a number of test distributions. We demonstrate the applicability of ISR in solving inverse problems, and compare it with INN on a benchmark inverse kinematics problem, as well as a geoacoustic inversion problem in oceanography (see (Chapman & Shang, 2021) for further information). Here, we aim to characterize the undersea environment, such as water-sediment depth, sound speed, etc., from acoustic signals. To the best of our knowledge, this work is the first attempt towards finding interpretable solutions to general nonlinear inverse problems by establishing analytical relationships between measurable quantities and unknown variables via symbolic invertible maps.

The remainder of the paper is organized as follows. In Section 2, we go through an overall background about Symbolic Regression (SR) and review the Equation Learner (EQL) network architecture. In Section 3, we introduce and present the proposed Invertible Symbolic Regression (ISR) method. We then show our results in Section 4, where we demonstrate the versatility of ISR as a density estimation method on a variety of examples (distributions), and then show its applicability in inverse problems on an inverse kinematics benchmark problem and through a case study in ocean geoacoustic inversion. Finally in Section 5, we provide our conclusions and outlook.

## 2 Background

Before diving into the proposed ISR method, we first delve into a comprehensive background of the Symbolic Regression (SR) task, as well as the Equation Learner (EQL) network architecture.

**Symbolic Regression.** Given a dataset $\mathcal{D} = \{\mathbf{x}_i, \mathbf{y}_i\}_{i=1}^N$ consisting of $N$ independent and identically distributed (i.i.d.) paired examples, where $\mathbf{x}_i \in \mathbb{R}^{d_\mathbf{x}}$ represents the input variables and $\mathbf{y}_i \in \mathbb{R}^{d_\mathbf{y}}$ the corresponding output for the $i$-th observation, the objective of SR is to find an analytical (symbolic) expression $f$ that best maps inputs to outputs, i.e. $\mathbf{y}_i \approx f(\mathbf{x}_i)$. SR seeks to identify the functional form of $f$ from the space of functions $\mathcal{S}$ defined by a set of given arithmetic operations (e.g. $+, -, \times, \div$) and mathematical functions (e.g. sin, cos, exp, etc.) that minimizes a predefined loss function $\mathcal{L}(f, \mathcal{D})$, which measures the discrepancy between the true outputs $\mathbf{y}_i$ and the predictions $f(\mathbf{x}_i)$ over all observations in the dataset. Unlike

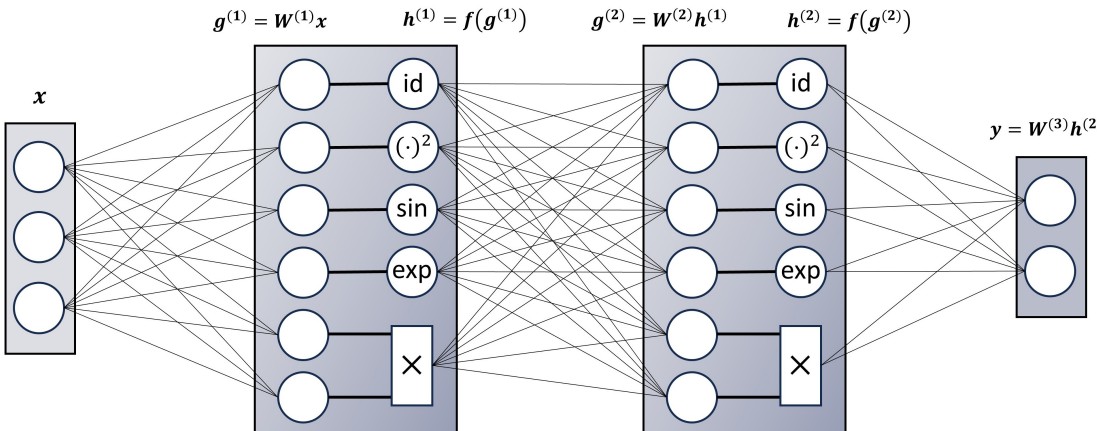

Figure 1: EQL network architecture for symbolic regression. For visual simplicity, we only show 2 hidden layers and 5 activation functions per layer (identity or "id", square, sine, exponential, and multiplication).

conventional regression methods that fit parameters within a predefined model structure, SR dynamically constructs the model structure itself, offering a powerful means to uncover underlying physical laws and/or nontrivial relationships.

**Equation Learner Network.** The Equation Learner (EQL) network is a multi-layer feed-forward neural network that is capable of performing symbolic regression by substituting traditional nonlinear activation functions with elementary functions. The EQL network was initially introduced by Martius & Lampert (2016) and Sahoo et al. (2018), and further explored by Kim et al. (2020). As shown in Figure 1, the EQL network architecture is based on a fully connected neural network where the ouput $\mathbf{h}^{(i)}$ of the $i$-th layer is given by

$$\mathbf{g}^{(i)} = \mathbf{W}^{(i)}\mathbf{h}^{(i-1)} \tag{1}$$

$$\mathbf{h}^{(i)} = f\big(\mathbf{g}^{(i)}\big) \tag{2}$$

where $\mathbf{W}^{(i)}$ is the weight matrix of the $i$-th layer, $f$ denotes the nonlinear (symbolic) activation functions, and $\mathbf{h}^{(0)} = \mathbf{x}$ represents the input data. In regression tasks, the final layer does not typically have an activation function, so the output for a network with $L$ hidden layers is given by

$$\mathbf{y} = \mathbf{h}^{(L+1)} = \mathbf{g}^{(L+1)} = \mathbf{W}^{(L+1)}\mathbf{h}^{(L)}. \tag{3}$$

In traditional neural networks, activation functions such as ReLU, tanh, or sigmoid are typically employed. However, for the EQL network, the activation function $f(\mathbf{g})$ may consist of a separate primitive function for each component of $\mathbf{g}$ (e.g. the square function, sine, exponential, etc.), and may include functions that take multiple arguments (e.g. the multiplication function). In addition, the primitive functions may be duplicated within each layer (to reduce the training's sensitivity to random initializations).

It is worth mentioning that, for visual simplicity, the schematic shown in Figure 1, shows an EQL network with only two hidden layers, where each layer has only five primitive functions, i.e., the activation function

$$f(\mathbf{g}) \in \big\{\text{identity, square, sine, exponential, multiplication}\big\}.$$

However, the EQL network can in fact include other functions or more hidden layers to fit a broader range (or class) of functions. Indeed, the number of hidden layers can dictate the complexity of the resulting symbolic expression and plays a similar role to the maximum depth of expression trees in genetic programming techniques. Although the EQL network may not offer the same level of generality as traditional symbolic regression methods, it is adequately capable of representing the majority of functions commonly encountered

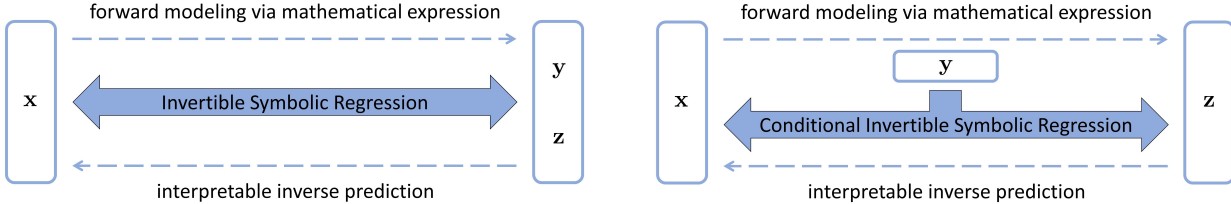

Figure 2: (Left) The proposed ISR framework learns a bijective symbolic transformation that maps the (unknown) variables $\mathbf{x}$ to the (observed) quantities $\mathbf{y}$ while transforming the lost information into latent variables $\mathbf{z}$. (Right) The conditional ISR (cISR) framework learns a bijective symbolic map that transforms $\mathbf{x}$ directly to a latent representation $\mathbf{z}$ given the observation $\mathbf{y}$. As we will show, both the forward and inverse mappings are efficiently computable and possess a tractable Jacobian, allowing explicit computation of posterior probabilities.

in scientific and engineering contexts. Crucially, the parametrized nature of the EQL network enables efficient optimization via gradient descent (and backpropagation).

After training the EQL network, the identified equation can be directly derived from the network weights. To avoid reaching overly complex symbolic expressions and to maintain interpretability, it is essential to guide the network towards learning the simplest expression that accurately represents the data. In methods based on genetic programming, this simplification is commonly achieved by restricting the number of terms in the expression. For the EQL network, this is attained by applying sparsity regularization to the network weights, which sets as many of these weights to zero as possible (e.g. $L_1$ regularization (Tibshirani, 1996), $L_{0.5}$ regularization (Xu et al., 2010)). In this work, we use a smoothed $L_{0.5}$ regularization (Wu et al., 2014; Fan et al., 2014), which was also adopted by Kim et al. (2020). Further details can be found in Appendix B.

## 3   Invertible Symbolic Regression

In this section, we delineate the problem setup for inverse problems, and then describe the proposed ISR approach.

### 3.1   Problem Specification

In various engineering and natural systems, the theories developed by experts describe how measurable (or observable) quantities $\mathbf{y} \in \mathbb{R}^{d_\mathbf{y}}$ result from the unknown (or hidden) properties $\mathbf{x} \in \mathbb{R}^{d_\mathbf{x}}$, known as the *forward process* $\mathbf{x} \to \mathbf{y}$. The goal of inverse prediction is to predict the unknown variables $\mathbf{x}$ from the observable variables $\mathbf{y}$, through the *inverse process* $\mathbf{y} \to \mathbf{x}$. As critical information is lost during the forward process (i.e. $d_\mathbf{x} \geq d_\mathbf{y}$), the inversion is usually intractable. Given that $f^{-1}(\mathbf{y})$ does not yield a uniquely defined solution, an effective inverse model should instead estimate the *posterior* probability distribution $p(\mathbf{x} \,|\, \mathbf{y})$ of the hidden variables $\mathbf{x}$, conditioned on the observed variable $\mathbf{y}$.

**Invertible Symbolic Regression (ISR).** Assume we are given a training dataset $\mathcal{D} = \{\mathbf{x}_i, \mathbf{y}_i\}_{i=1}^N$, collected using forward model $\mathbf{y} = s(\mathbf{x})$ and prior $p(\mathbf{x})$. To counteract the loss of information during the forward process, we introduce latent random variables $\mathbf{z} \in \mathbb{R}^{d_\mathbf{z}}$ drawn from a multivariate standard normal distribution, i.e. $\mathbf{z} \sim p_\mathbf{z}(\mathbf{z}) = \mathcal{N}(\mathbf{0}, \boldsymbol{I}_{d_\mathbf{z}})$, where $d_\mathbf{z} = d_\mathbf{x} - d_\mathbf{y}$. These latent variables are designed to capture the information related to $\mathbf{x}$ that is not contained in $\mathbf{y}$ (Ardizzone et al., 2019a). In ISR, we aim to learn a bijective symbolic function $f : \mathbb{R}^{d_\mathbf{x}} \to \mathbb{R}^{d_\mathbf{y}} \times \mathbb{R}^{d_\mathbf{z}}$ from the space of functions defined by a set of mathematical functions (e.g. sin, cos, exp, log) and arithmetic operations (e.g. $+, -, \times, \div$), and such that

$$[\mathbf{y}, \mathbf{z}] = f(\mathbf{x}) = \big[ f_\mathbf{y}(\mathbf{x}), f_\mathbf{z}(\mathbf{x}) \big], \qquad \mathbf{x} = f^{-1}(\mathbf{y}, \mathbf{z}) \tag{4}$$

where $f_\mathbf{y}(\mathbf{x}) \approx s(\mathbf{x})$ is an approximation of the forward process $s(x)$. As discussed later, we will learn $f$ (and hence $f^{-1}$) through an invertible symbolic architecture with bi-directional training. The solution of the

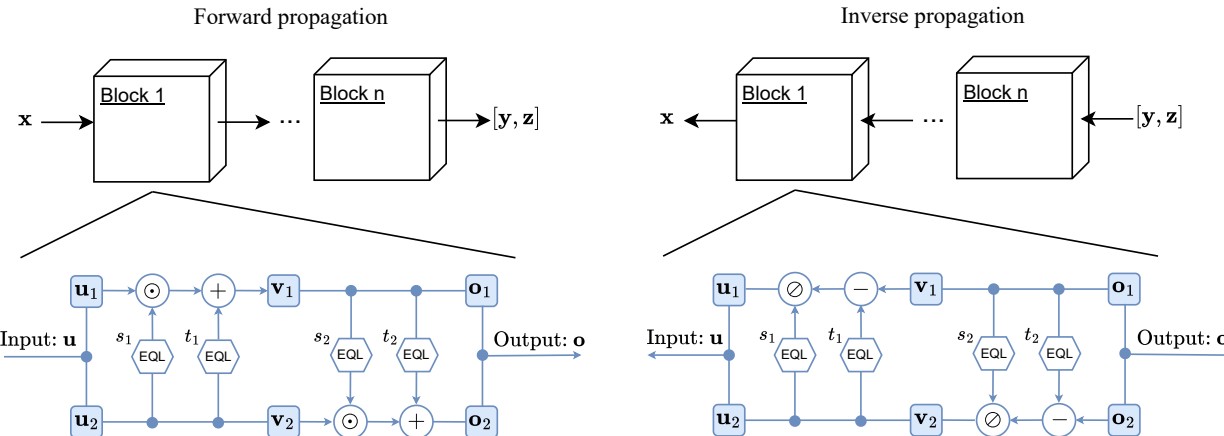

Figure 3: The proposed ISR method integrates EQL within the affine coupling blocks of the INN invertible architecture.[1] This results in a bijective symbolic transformation that is both easily invertible and has a tractable Jacobian. Indeed, the forward and inverse directions both possess identical computational cost. Here, $\odot$ and $\oslash$ denote element-wise multiplication and divison, respectively.

inverse problem (i.e. the posterior $p(\mathbf{x} \,|\, \mathbf{y}^*)$) can then be found by calling $f^{-1}$ for a fixed observation $\mathbf{y}^*$ while randomly (and repeatedly) sampling the latent variable $\mathbf{z}$ from the same standard Gaussian distribution.

**Conditional Invertible Symbolic Regression (cISR).** Inspired by works on conditional invertible neural networks (cINNs) (Ardizzone et al., 2019b; 2021; Kruse et al., 2021; Luce et al., 2023), instead of training ISR to predict $\mathbf{y}$ from $\mathbf{x}$ while transforming the lost information into latent variables $\mathbf{z}$, we train them to transform $\mathbf{x}$ directly to latent variables $\mathbf{z}$ given the observed variables $\mathbf{y}$. This is achieved by incorporating $\mathbf{y}$ as an additional input within the bijective symbolic architecture during both the forward and inverse passes (see Figure 2). cISR works with larger latent spaces than ISR since $d_{\mathbf{z}} = d_{\mathbf{x}}$ regardless of the dimension $d_{\mathbf{y}}$ of the observed quantities $\mathbf{y}$. Further details are provided in the following section.

In addition to approximating the forward model via mathematical relations, ISR also identifies an interpretable inverse map via analytical expressions (see Figure 2). Such interpretable mappings are of particular interest in physical sciences, where an ambitious objective involves creating intelligent machines capable of generating novel scientific findings(Udrescu & Tegmark, 2020; Udrescu et al., 2020; Liu & Tegmark, 2021; Keren et al., 2023; Liu et al., 2024). As described next, the ISR architecture is both easily invertible and has a tractable Jacobian, allowing for explicit computation of posterior probabilities.

## 3.2 Invertible Symbolic Architecture

The general SR problem is to search the space of functions to find the optimal analytical expression given data. As discussed in Section 3.1, the objective of ISR is to analytically learn a bijection (more specifically a diffeomorphism (Teshima et al., 2020)) via symbolic distillation. In other words, the objective of ISR is essentially similar to that of the general SR problem with the additional constraint that the resulting model has to be invertible. Traditionally, many SR methods rely on Genetic Programming (Tohme et al., 2023a) to search for the optimal symbolic expression. While various methods can be used to learn the bijective symbolic function $f$ in Eq. (4), as discussed next, we resort to coupling-based invertible architectures coupled with EQL networks, whose parametrized nature enhances computational efficiency.

Inspired by the architectures proposed by Dinh et al. (2016); Kingma & Dhariwal (2018); Ardizzone et al. (2019a), we adopt a fully invertible architecture mainly defined by a sequence of $n$ reversible blocks where each block consists of two complementary affine coupling layers. In particular, we first split the block's input $\mathbf{u} \in \mathbb{R}^{d_{\mathbf{u}}}$ into $\mathbf{u}_1 \in \mathbb{R}^{d_{\mathbf{u}_1}}$ and $\mathbf{u}_2 \in \mathbb{R}^{d_{\mathbf{u}_2}}$ (where $d_{\mathbf{u}_1} + d_{\mathbf{u}_2} = d_{\mathbf{u}}$), which are fed into the coupling layers as

follows:

$$\begin{bmatrix} \mathbf{v}_1 \\ \mathbf{v}_2 \end{bmatrix} = \begin{bmatrix} \mathbf{u}_1 \odot \exp\left(s_1(\mathbf{u}_2)\right) + t_1(\mathbf{u}_2) \\ \mathbf{u}_2 \end{bmatrix}, \qquad \begin{bmatrix} \mathbf{o}_1 \\ \mathbf{o}_2 \end{bmatrix} = \begin{bmatrix} \mathbf{v}_1 \\ \mathbf{v}_2 \odot \exp\left(s_2(\mathbf{v}_1)\right) + t_2(\mathbf{v}_1) \end{bmatrix}, \qquad (5)$$

where $\odot$ denotes the Hadamard product or element-wise multiplication. The outputs $[\mathbf{o}_1, \mathbf{o}_2]$ are then concatenated again and passed to the next coupling block. The internal mappings $s_1$ and $t_1$ are functions from $\mathbb{R}^{d_{\mathbf{u}_2}} \to \mathbb{R}^{d_{\mathbf{u}_1}}$, and $s_2$ and $t_2$ are functions from $\mathbb{R}^{d_{\mathbf{u}_1}} \to \mathbb{R}^{d_{\mathbf{u}_2}}$. In general, $s_i$ and $t_i$ can be arbitrarily complicated functions (e.g. neural networks as in Ardizzone et al. (2019a)). In our proposed ISR approach, they are represented by EQL networks (see Figure 3), resulting in a fully symbolic invertible architecture. Moving forward, we shall refer to them as EQL *subnetworks* of the block.

The transformations above result in upper and lower triangular Jacobians:

$$J_{\mathbf{u} \mapsto \mathbf{v}} = \begin{bmatrix} \mathrm{diag}\left(\exp\left(s_1(\mathbf{u}_2)\right)\right) & \frac{\partial \mathbf{v}_1}{\partial \mathbf{u}_2} \\ 0 & I \end{bmatrix}, \qquad J_{\mathbf{v} \mapsto \mathbf{o}} = \begin{bmatrix} I & 0 \\ \frac{\partial \mathbf{o}_2}{\partial \mathbf{v}_1} & \mathrm{diag}\left(\exp\left(s_2(\mathbf{v}_1)\right)\right) \end{bmatrix}. \qquad (6)$$

Hence, their determinants can be trivially computed:

$$\det\left(J_{\mathbf{u} \mapsto \mathbf{v}}\right) = \prod_{i=1}^{d_{\mathbf{u}_1}} \exp\left([s_1(\mathbf{u}_2)]_i\right) = \exp\left(\sum_{i=1}^{d_{\mathbf{u}_1}} [s_1(\mathbf{u}_2)]_i\right),$$

$$\det\left(J_{\mathbf{v} \mapsto \mathbf{o}}\right) = \prod_{i=1}^{d_{\mathbf{u}_2}} \exp\left([s_2(\mathbf{v}_1)]_i\right) = \exp\left(\sum_{i=1}^{d_{\mathbf{u}_2}} [s_2(\mathbf{v}_1)]_i\right). \qquad (7)$$

Then, the resulting Jacobian determinant of the coupling block is given by

$$\begin{aligned} \det\left(J_{\mathbf{u} \mapsto \mathbf{o}}\right) &= \det\left(J_{\mathbf{u} \mapsto \mathbf{v}}\right) \cdot \det\left(J_{\mathbf{v} \mapsto \mathbf{o}}\right) \\ &= \exp\left(\sum_{i=1}^{d_{\mathbf{u}_1}} [s_1(\mathbf{u}_2)]_i\right) \cdot \exp\left(\sum_{i=1}^{d_{\mathbf{u}_2}} [s_2(\mathbf{v}_1)]_i\right) \\ &= \exp\left(\sum_{i=1}^{d_{\mathbf{u}_1}} [s_1(\mathbf{u}_2)]_i + \sum_{i=1}^{d_{\mathbf{u}_2}} [s_2(\mathbf{v}_1)]_i\right) \\ &= \exp\left(\sum_{i=1}^{d_{\mathbf{u}_1}} [s_1(\mathbf{u}_2)]_i + \sum_{i=1}^{d_{\mathbf{u}_2}} \left[s_2\left(\mathbf{u}_1 \odot \exp\left(s_1(\mathbf{u}_2)\right) + t_1(\mathbf{u}_2)\right)\right]_i\right) \end{aligned} \qquad (8)$$

which can be efficiently calculated. Indeed, the Jacobian determinant of the whole map $\mathbf{x} \to [\mathbf{y}, \mathbf{z}]$ is the product of the Jacobian determinants of the $n$ underlying coupling blocks (see Figure 3).

Given the output $\mathbf{o} = [\mathbf{o}_1, \mathbf{o}_2]$, the expressions in Eqs. (5) are clearly invertible:

$$\mathbf{u}_2 = \left(\mathbf{o}_2 - t_2(\mathbf{o}_1)\right) \oslash \exp\left(s_2(\mathbf{o}_1)\right), \qquad \mathbf{u}_1 = \left(\mathbf{o}_1 - t_1(\mathbf{u}_2)\right) \oslash \exp\left(s_1(\mathbf{u}_2)\right) \qquad (9)$$

where $\oslash$ denotes element-wise division. Crucially, even when the coupling block is inverted, the EQL subnetworks $s_i$ and $t_i$ need *not* themselves be invertible; they are only ever evaluated in the forward direction. We denote the whole ISR map $\mathbf{x} \to [\mathbf{y}, \mathbf{z}]$ as $f(\mathbf{x}; \theta) = \left[f_{\mathbf{y}}(\mathbf{x}; \theta), f_{\mathbf{z}}(\mathbf{x}; \theta)\right]$ parameterized by the EQL subnetworks parameters $\theta$, and the inverse as $f^{-1}(\mathbf{y}, \mathbf{z}; \theta)$.

*Remark* 3.1. The proposed ISR architecture consists of a sequence of these symbolic reversible blocks. To enhance the model's predictive and expressive capability, we can: i) increase the number of reversible coupling blocks, ii) increase the number of hidden layers in each underlying EQL network, iii) increase the number of hidden neurons per layer in each underlying EQL network, or iv) increase the complexity of the symbolic activation functions used in the EQL network. However, it is worth noting that these enhancements come with a trade-off, as they inevitably lead to a decrease in the model's interpretability.

*Remark* 3.2. To further improve the model capacity, as in Ardizzone et al. (2019a), we incorporate (random, but fixed) permutation layers between the coupling blocks, which shuffles the input elements for subsequent coupling blocks. This effectively randomizes the configuration of splits $\mathbf{u} = [\mathbf{u}_1, \mathbf{u}_2]$ across different blocks, thereby enhancing interplay between variables.

---

[1]As direct division can lead to numerical issues, we apply the exponential function to $s_i$ (after clipping its extreme values) in the formulation described in Eq. (5). This also guarantees non-zero diagonal entries in the Jacobian matrices.

*Remark* 3.3. Inspired by Ardizzone et al. (2019a), we split the coupling block's input vector $\mathbf{u} \in \mathbb{R}^{d_{\mathbf{u}}}$ into two halves, i.e. $\mathbf{u}_1 \in \mathbb{R}^{d_{\mathbf{u}_1}}$ and $\mathbf{u}_2 \in \mathbb{R}^{d_{\mathbf{u}_2}}$ where $d_{\mathbf{u}_1} = \lfloor \frac{d_{\mathbf{u}}}{2} \rfloor$ and $d_{\mathbf{u}_2} = d_{\mathbf{u}} - d_{\mathbf{u}_1}$. In the case where $\mathbf{u}$ is one-dimensional (or scalar), i.e. $d_{\mathbf{u}} = 1$ and $\mathbf{u} \in \mathbb{R}$, we pad it with an extra zero (so that $d_{\mathbf{u}} = 2$) along with a loss term that prevents the encoding of information in the extra dimension (e.g. we use the $L_2$ loss to maintain those values near zero).

*Remark* 3.4. The proposed ISR architecture is also compatible with the conditional ISR (cISR) framework proposed in the previous section. In essence, cISR identifies a bijective symbolic transformation directly between $\mathbf{x}$ and $\mathbf{z}$ given the observation $\mathbf{y}$. This is attained by feeding $\mathbf{y}$ as an extra input to each coupling block, during both the forward and inverse passes. In particular, and as suggested by Ardizzone et al. (2019b; 2021); Kruse et al. (2021), we adapt the same coupling layers given by Eqs. (5) and (9) to produce a conditional coupling block. Since the subnetworks $s_i$ and $t_i$ are never inverted, we enforce the condition on the observation by concatenating $\mathbf{y}$ to their inputs without losing the invertibility, i.e. we replace $s_1(\mathbf{u}_2)$ with $s_1(\mathbf{u}_2, \mathbf{y})$, etc. In complex settings, the condition $\mathbf{y}$ is first fed into a separate feed-forward conditioning network, resulting in higher-level conditioning features that are then injected into the conditional coupling blocks. Although cISR can have better generative properties (Kruse et al., 2021), it leads to more complex symbolic expressions and less interpretability as it explicitly conditions the map on the observation within the symbolic formulation. We denote the entire cISR forward map $\mathbf{x} \to \mathbf{z}$ as $f(\mathbf{x}; \mathbf{y}, \theta)$ parameterized by $\theta$, and the inverse as $f^{-1}(\mathbf{z}; \mathbf{y}, \theta)$.

### 3.3 Maximum Likelihood Training of ISR

We train the proposed ISR model to learn a bijective symbolic transformation $f : \mathbb{R}^{d_{\mathbf{x}}} \to \mathbb{R}^{d_{\mathbf{y}}} \times \mathbb{R}^{d_{\mathbf{z}}}$. There are various choices to define the loss functions with different advantage and disadvantages (Grover et al., 2018; Ren et al., 2020; Ardizzone et al., 2019a; 2021; Kruse et al., 2021). As reported in Kruse et al. (2021), there are two main training approaches:

i) A standard supervised $L_2$ loss for fitting the model's $\mathbf{y}$ predictions to the training data, combined with a Maximum Mean Discrepancy (MMD) (Gretton et al., 2012; Ardizzone et al., 2019a) for fitting the latent distribution $p_{\mathbf{z}}(\mathbf{z})$ to $\mathcal{N}(\mathbf{0}, \boldsymbol{I}_{d_{\mathbf{z}}})$, given samples.

ii) A Maximum Likelihood Estimate (MLE) loss that enforces $\mathbf{z}$ to be standard Gaussian, i.e. $\mathbf{z} \sim p_{\mathbf{z}}(\mathbf{z}) = \mathcal{N}(\mathbf{0}, \boldsymbol{I}_{d_{\mathbf{z}}})$ and by approximating the distribution on $\mathbf{y}$ with a Gaussian distribution around the ground truth values $\mathbf{y}_{\text{gt}}$ with very low variance $\sigma^2$ (Dinh et al., 2016; Ren et al., 2020; Kruse et al., 2021).

Given that MLE is shown to perform well as reported in the literature (Ardizzone et al., 2019a), we apply it here. Next, we demonstrate how this approach is equivalent to minimizing the forward Kullback-Leibler (KL) divergence as the cost (cf. (Papamakarios et al., 2021)). We note that given the map $f(\mathbf{x}; \theta) \mapsto [\mathbf{z}, \mathbf{y}]$, parameterized by $\theta$, and assuming $\mathbf{y}$ and $\mathbf{z}$ are independent, the density $p_{\mathbf{x}}$ relates to $p_{\mathbf{Y}}$ and $p_{\mathbf{z}}$ through the change-of-variables formula

$$p_{\mathbf{x}}(\mathbf{x}; \theta) = p_{\mathbf{Y}}\big(\mathbf{y} = f_{\mathbf{y}}(\mathbf{x}; \theta)\big) \, p_{\mathbf{z}}\big(\mathbf{z} = f_{\mathbf{z}}(\mathbf{x}; \theta)\big) \cdot \big|\det\big(J_{\mathbf{x} \mapsto [\mathbf{z}, \mathbf{y}]}(\mathbf{x}; \theta)\big)\big|. \tag{10}$$

where $J_{\mathbf{x} \mapsto [\mathbf{z}, \mathbf{y}]}(\mathbf{x}; \theta)$ denotes the Jacobian of the map $f$ parameterized by $\theta$. This expression is then used to define the loss function, which we derive by following the work in (Papamakarios et al., 2021). In particular, we aim to minimize the forward KL divergence between a target distribution $p_{\mathbf{x}}^*(\mathbf{x})$ and our *flow-based* model $p_{\mathbf{x}}(\mathbf{x}; \theta)$, given by

$$\begin{aligned}
\mathcal{L}(\theta) &= D_{\text{KL}}\big[p_{\mathbf{x}}^*(\mathbf{x}) \,\big|\big|\, p_{\mathbf{x}}(\mathbf{x}; \theta)\big] \\
&= -\mathbb{E}_{p_{\mathbf{x}}^*(\mathbf{x})}\big[\log p_{\mathbf{x}}(\mathbf{x}; \theta)\big] + \text{const.} \\
&= -\mathbb{E}_{p_{\mathbf{x}}^*(\mathbf{x})}\big[\log p_{\mathbf{Y}}\big(f_{\mathbf{y}}(\mathbf{x}; \theta)\big) + \log p_{\mathbf{z}}\big(f_{\mathbf{z}}(\mathbf{x}; \theta)\big) + \log\big|\det\big(J_{\mathbf{x} \mapsto [\mathbf{z}, \mathbf{y}]}(\mathbf{x}; \theta)\big)\big|\big] + \text{const.}
\end{aligned} \tag{11}$$

The forward KL divergence is particularly suitable for cases where we have access to samples from the target distribution, but we cannot necessarily evaluate the target density $p_{\mathbf{x}}^*(\mathbf{x})$. Assuming we have a set of samples $\{\mathbf{x}_i\}_{i=1}^N$ from $p_{\mathbf{x}}^*(\mathbf{x})$, we can approximate the expectation in Eq. (11) using Monte Carlo integration as

$$\mathcal{L}(\theta) \approx -\frac{1}{N} \sum_{i=1}^N \Big( \log p_{\mathbf{Y}}\big(f_{\mathbf{y}}(\mathbf{x}_i; \theta)\big) + \log p_{\mathbf{z}}\big(f_{\mathbf{z}}(\mathbf{x}_i; \theta)\big) + \log\big|\det\big(J_{\mathbf{x} \mapsto [\mathbf{z}, \mathbf{y}]}(\mathbf{x}_i; \theta)\big)\big| + \text{const.} \Big) . \tag{12}$$

As we can see, minimizing the above Monte Carlo approximation of the KL divergence is equivalent to maximizing likelihood (or minimizing negative log-likelihood). Assuming $p_{\mathbf{z}}$ is standard Gaussian and $p_{\mathbf{Y}}$ is a multivariate normal distribution around $\mathbf{y}_{\text{gt}}$, the negative log-likelihood (NLL) loss in Eq. (12) becomes

$$\mathcal{L}_{\text{NLL}}(\theta) = \frac{1}{N} \sum_{i=1}^{N} \left( \frac{1}{2} \cdot \frac{\left( f_{\mathbf{y}}(\mathbf{x}_i; \theta) - \mathbf{y}_{\text{gt}} \right)^2}{\sigma^2} + \frac{1}{2} \cdot f_{\mathbf{z}}(\mathbf{x}_i; \theta)^2 - \log \left| \det \left( J_{\mathbf{x} \mapsto [\mathbf{z}, \mathbf{y}]}(\mathbf{x}_i; \theta) \right) \right| \right) . \tag{13}$$

In other words, we find the optimal ISR parameters $\theta$ by minimizing the NLL loss in Eq. (13), and the resulting bijective symbolic expression can be directly extracted from the these optimal parameters.

*Remark* 3.5. We note that cISR is also suited for maximum likelihood training. Given the conditioning observation $\mathbf{y}$, the density $p_{\mathbf{x} \mid \mathbf{Y}}$ relates to $p_{\mathbf{z}}$ through the change-of-variables formula

$$p_{\mathbf{x} \mid \mathbf{Y}}(\mathbf{x} \mid \mathbf{y}, \theta) = p_{\mathbf{z}}\big( \mathbf{z} = f(\mathbf{x}; \mathbf{y}, \theta) \big) \cdot \left| \det \big( J_{\mathbf{x} \mapsto \mathbf{z}}(\mathbf{x}; \mathbf{y}, \theta) \big) \right|, \tag{14}$$

where $J_{\mathbf{x} \mapsto \mathbf{z}}(\mathbf{x}; \mathbf{y}, \theta)$ indicates the Jacobian of the map $f$ conditioned on $\mathbf{y}$ and parameterized by $\theta$. Following the same procedure as above, the cISR model can be trained by minimizing the following NLL loss function

$$\mathcal{L}_{\text{NLL}}(\theta) = \frac{1}{N} \sum_{i=1}^{N} \left( \frac{1}{2} \cdot f(\mathbf{x}_i; \mathbf{y}_i, \theta)^2 - \log \left| \det \big( J_{\mathbf{x} \mapsto \mathbf{z}}(\mathbf{x}_i; \mathbf{y}_i, \theta) \big) \right| \right) . \tag{15}$$

As we will show in the next section, if we ignore the condition on the observation $\mathbf{y}$, the loss in Eq. 15 can also be used for training ISR as a normalizing flow for the unsupervised learning task of approximating a target probability density function from samples (cf. Eq. 18).

## 4 Results

We evaluate our proposed ISR method on a variety of problems. We first show how ISR can serve as a normalizing flow for density estimation tasks on several test distributions. We then demonstrate the capabilities of ISR in solving inverse problems by considering two synthetic problems and then a more challenging application in ocean acoustics (Jensen et al., 2011; Ali et al., 2023; Huang et al., 2006; Dosso & Dettmer, 2011; Bianco et al., 2019; Holland et al., 2005; Benson et al., 2000). We mainly compare our ISR approach against INN (Ardizzone et al., 2019a) throughout our experiments. Further experimental details can be found in Appendix B.

### 4.1 Leveraging ISR for Density Estimation via Normalizing Flow

Given $N$ independently and identically distributed (i.i.d.) samples, i.e. $\{\mathbf{X}_i\}_{i=1}^{N} \sim p_{\mathbf{x}}^{\text{target}}$, we would like to estimate the target density $p_{\mathbf{x}}^{\text{target}}$ and generate new samples from it. This problem categorizes as the density estimation where non-parametric, e.g. Kernel Density Estimation (Sheather, 2004), and parametric estimators, e.g. Maximum Entropy Distribution as the least biased estimator (Tohme et al., 2024), are classically used. In recent years, this problem has been approached using normalizing flow equipped with invertible map, which has gained a great deal of interest in the generative AI task. In an attempt to introduce interpretability in the trained model, we extend invertible normalizing flow to symbolic framework using the proposed ISR architecture.

In order to use the proposed cISR method as the normalizing flow for the unsupervised task of resampling from an intractable target distribution, we drop out $\mathbf{y}$ and enforce $d_{\mathbf{x}} = d_{\mathbf{z}}$.[2] In this case, we aim to learn a an invertible and symbolic map $f : \mathbb{R}^{d_{\mathbf{x}}} \to \mathbb{R}^{d_{\mathbf{z}}}$, parameterized by $\theta$ such that

$$\mathbf{z} = f(\mathbf{x}; \theta), \qquad \mathbf{x} = f^{-1}(\mathbf{z}; \theta), \tag{16}$$

---

[2]In the absence of $\mathbf{y}$, cISR and ISR are equivalent, so we simply refer to them as ISR. Similarly, INN and cINN become equivalent, and we simply refer to them as INN.

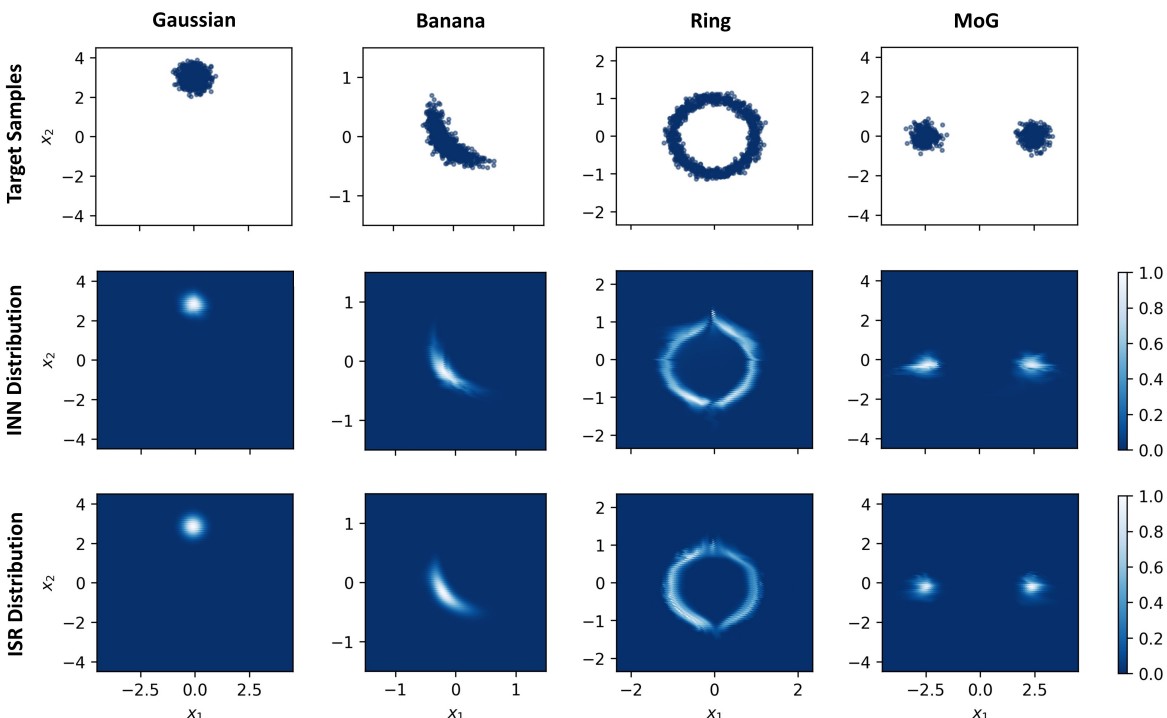

Figure 4: Samples from four different target densities (first row), and their estimated distributions using INN (second row) and the proposed ISR method (third row).

where $\mathbf{z} \sim p_{\mathbf{z}}$ is the standard normal distribution function, which is easy to sample from. Using the change-of-variables formula, the density $p_{\mathbf{x}}$ relates to the density $p_{\mathbf{z}}$ via

$$p_{\mathbf{x}}(\mathbf{x}; \theta) = p_{\mathbf{z}}\big(\mathbf{z} = f(\mathbf{x}; \theta)\big) \cdot \big|\det\big(J_{\mathbf{x} \mapsto \mathbf{z}}(\mathbf{x}; \theta)\big)\big|, \tag{17}$$

where $J_{\mathbf{x} \mapsto \mathbf{z}}$ indicates the Jacobian of the map $f$ parameterized by $\theta$. Following the same procedure outlined in Section 3.3, the model can be trained by minimizing the following NLL loss function

$$\mathcal{L}_{\mathrm{NLL}}(\theta) = \frac{1}{N} \sum_{i=1}^{N} \left( \frac{1}{2} \cdot f(\mathbf{x}_i; \theta)^2 - \log \big|\det\big(J_{\mathbf{x} \mapsto \mathbf{z}}(\mathbf{x}_i; \theta)\big)\big| \right) . \tag{18}$$

We compare the proposed ISR approach with INN in recovering several two-dimensional target distributions (i.e. $d_{\mathbf{x}} = d_{\mathbf{z}} = 2$). First, we consider a fairly simple multivariate normal distribution $\mathcal{N}\big(\boldsymbol{\mu}, \boldsymbol{\Sigma}\big)$ with mean $\boldsymbol{\mu} = [0, 3]$ and covariance matrix $\boldsymbol{\Sigma} = \frac{1}{10} \cdot \boldsymbol{I}_2$ as the target density. Then, we consider more challenging distributions: the "Banana," "Mixture of Gaussians (MoG)," and "Ring" distributions that are also considered in Jaini et al. (2019); Wenliang et al. (2019). For each of these target distributions, we draw $N_s = 10^4$ i.i.d. samples and train an invertible map that transport the samples to a standard normal distribution. This is called normalizing flow, where we intend to compare the standard INN with the proposed ISR architecture. Here, we use a single coupling block for the Gaussian and banana cases, and two coupling blocks for the ring and MoG test cases.

As shown in Figure 4, the proposed ISR method finds and generates samples of the considered target densities with slightly better accuracy than INN. We report the bijective symbolic expressions in Table 1 of Appendix A. For instance, the first target distribution in Figure 4 is the two-dimensional multivariate Gaussian distribution $\mathcal{N}\big(\boldsymbol{\mu}, \boldsymbol{\Sigma}\big)$ with mean $\boldsymbol{\mu} = [0, 3]$ and covariance matrix $\boldsymbol{\Sigma} = \frac{1}{10} \cdot \boldsymbol{I}_2$. This is indeed a shifted and scaled standard Gaussian distribution where we know the analytical solution to the true map:

$$\mathbf{X} = \begin{bmatrix} X_1 \\ X_2 \end{bmatrix} \sim \mathcal{N}\left(\boldsymbol{\mu}, \frac{1}{10} \cdot \boldsymbol{I}_2\right) \sim \boldsymbol{\mu} + \sqrt{\frac{1}{10}} \cdot \mathcal{N}\left(\mathbf{0}, \boldsymbol{I}_2\right)$$

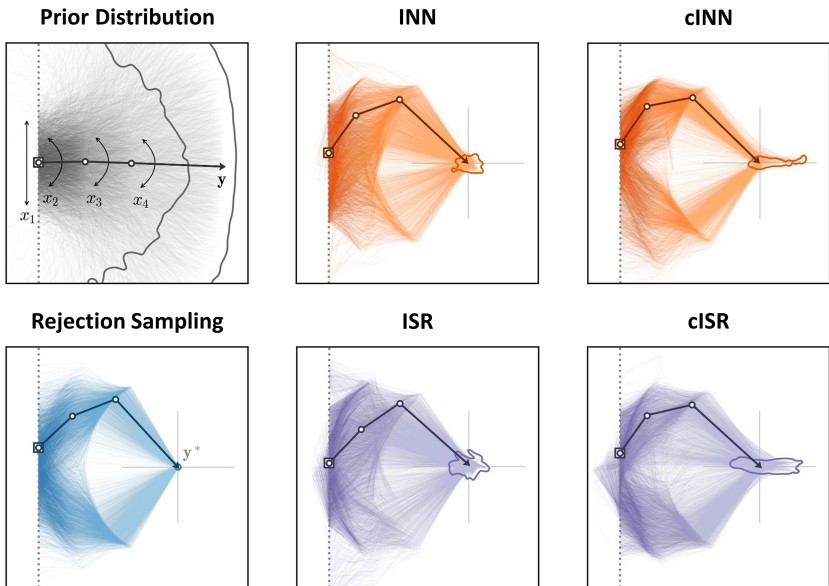

Figure 5: Results for the inverse kinematics benchmark problem. The faint colored lines indicate sampled arm configurations $\mathbf{x}$ taken from each model's predicted posterior $\hat{p}(\mathbf{x} \mid \mathbf{y}^*)$, conditioned on the target end point $\mathbf{y}^*$, which is indicated by a gray cross. The contour lines around the target end point enclose the regions containing 97% of the sampled arms' end points. We emphasize the arm with the highest estimated likelihood as a bold line.

$$= \boldsymbol{\mu} + \frac{1}{\sqrt{10}} \cdot \mathbf{Z} = \begin{bmatrix} 0 \\ 3 \end{bmatrix} + 0.316 \cdot \begin{bmatrix} Z_1 \\ Z_2 \end{bmatrix} = \begin{bmatrix} 0.316\, Z_1 \\ 3 + 0.316\, Z_2 \end{bmatrix} . \tag{19}$$

As shown in Table 1 of Appendix A, for this Gaussian distribution example, the proposed ISR method finds the following invertible expression:

$$z_1 = x_1 \cdot e^{1.16} = 3.19\, x_1 \qquad \Longleftrightarrow \qquad x_1 = 3.19^{-1}\, z_1 = 0.313\, z_1$$
$$z_2 = x_2 \cdot e^{1.14} - 9.39 = 3.13\, x_2 - 9.39 \qquad \Longleftrightarrow \qquad x_2 = 3.13^{-1}\, (z_2 + 9.39) = 0.319\, z_2 + 3.00$$

In other words, the proposed ISR method identifies the true underlying transformation given by Eq. (19) with a high accuracy. As discussed in Appendix B. However, the user can indeed add other operators (e.g. log, etc.) when necessary or when domain knowledge is available,

Appendix D explores a more challenging and noteworthy example through a toy inverse problem.

## 4.2 Inverse Kinematics

We now consider a geometrical benchmark example used by Ardizzone et al. (2019a); Kruse et al. (2021), which simulates an inverse kinematics problem in a two-dimensional space: A multi-jointed 2D arm moves vertically along a rail and rotates at three joints. In this problem, we are interested in the configurations (i.e. the four degrees of freedom) of the arm that place the arm's end point at a given position. The forward process computes the coordinates of the end point $\mathbf{y} \in \mathbf{R}^2$, given a configuration $\mathbf{x} \in \mathbf{R}^4$ (i.e. $d_{\mathbf{x}} = 4$, $d_{\mathbf{y}} = 2$, and hence $d_{\mathbf{z}} = 2$). In particular, the forward process takes $\mathbf{x} = [x_1, x_2, x_3, x_4]$ as argument, where $x_1$ denotes the arm's starting height, and $x_2$, $x_3$, $x_4$ are its three joint angles, and returns the coordinates of its end point $\mathbf{y} = [y_1, y_2]$ given by

$$y_1 = \ell_1 \sin(x_2) + \ell_2 \sin(x_2 + x_3) + \ell_3 \sin(x_2 + x_3 + x_4) + x_1$$
$$y_2 = \ell_1 \cos(x_2) + \ell_2 \cos(x_2 + x_3) + \ell_3 \cos(x_2 + x_3 + x_4) \tag{20}$$

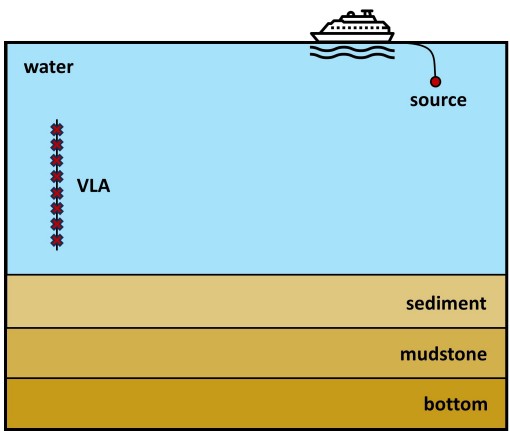

Figure 6: The SWellEx-96 experiment environment. The acoustic source is towed by a research vessel and transmits signals at various frequencies. The acoustic sensor consists of a vertical line array (VLA). Based on the measurements collected at the VLA, the objective is to estimate posterior distributions over parameters of interest (e.g. water depth, sound speed at the water-sediment interface, source range and depth, etc.).

where the segment lengths $\ell_1 = 0.5$, $\ell_2 = 0.5$, and $\ell_3 = 1$. The parameters $\mathbf{x}$ follow a Gaussian prior $\mathbf{x} \sim \mathcal{N}(\mathbf{0}, \boldsymbol{\sigma}^2 \cdot \boldsymbol{I}_4)$ with $\boldsymbol{\sigma}^2 = [0.25^2, 0.25, 0.25, 0.25]$, which favors a configuration with a centered origin and $180°$ joint angles (see Figure 5). We consider a training dataset of size $10^6$, constructed using this Gaussian prior and the forward process in Eq. (20). The inverse problem here asks to find the posterior distribution $p(\mathbf{x} \mid \mathbf{y}^*)$ of all possible configurations (or parameters) $\mathbf{x}$ that result in the arm's end point being positioned at a given $\mathbf{y}^*$ location. This inverse kinematics problem, being low-dimensional, offers computationally inexpensive forward (and backward) process, which enables fast training, intuitive visualizations, and an approximation of the true posterior estimates via rejection sampling.[3]

An example of a challenging end point $\mathbf{y}^*$ is shown in Figure 5, where we compare the proposed ISR method against the approximate true posterior (obtained via rejection sampling), as well as INN. The chosen $\mathbf{y}^*$ is particularly challenging, since this end point is unlikely under the prior $p(\mathbf{x})$, and results in a strongly bi-modal posterior $p(\mathbf{x} \mid \mathbf{y}^*)$ (Ardizzone et al., 2019a; Kruse et al., 2021). As we can observe in Figure 5, compared to rejection sampling, all the considered architectures (i.e. INN, cINN, ISR, and cISR) are able to capture the two symmetric modes well. However, we can clearly see that they all generate $\mathbf{x}$-samples such that their resulting end points miss the target $\mathbf{y}^*$ by a wider margin. Quantitative results are also provided in Appendix C.

### 4.3 Application: Geoacoustic Inversion

Predicting acoustic propagation at sea is vital for various applications, including sonar performance forecasting and mitigating noise pollution at sea. The ability to predict sound propagation in a shallow water environment depends on understanding the seabed's geoacoustic characteristics. Inferring those characteristics from ocean acoustic measurements (or signals) is known as geoacoustic inversion (GI). GI involves several components: (i) representation of the ocean environment, (ii) selection of the inversion method, including the forward propagation model implemented, and (iii) quantification of the uncertainty related to the parameters estimates.

---

[3]**Rejection sampling.** Assume we require $N_s$ samples of $\mathbf{x}$ from the posterior $p(\mathbf{x} \mid \mathbf{y}^*)$ given some observation $\mathbf{y}^*$. After setting some acceptance threshold $\epsilon$, we iteratively generate $\mathbf{x}$-samples from the prior. For each sample, we simulate the corresponding $\mathbf{y}$-values and only keep those with $\text{dist}(\mathbf{y}, \mathbf{y}^*) < \epsilon$. The process is repeated until $N_s$ samples are collected (or accepted). Indeed, the smaller the threshold $\epsilon$, the more $\mathbf{x}$-samples candidates (and hence the more simulations) have to be generated. Hence, we adopt this approach in this low-dimensional inverse kinematics problem, where we can afford to run the forward process (or simulation) a huge number of times.

We start by describing the ocean environment. We consider the setup of SWellEx-96 Yardim et al. (2010); Meyer & Gemba (2021), which was an experiment done off the coast of San Diego, CA, near Point Loma. This experimental setting is one of the most used, documented, and understood studies in the undersea acoustics community.[4] As depicted in Figure 6, the data is collected via a vertical line array (VLA). The specification of the 21 hydrophones of the VLA and sound speed profile (SSP) in the water column is provided in the SWellEx-96 documentation. The SSP and sediment parameters are considered to be range-independent. Water depth refers to the depth of the water at the array. The source is towed by a research vessel which consists of a comb signal comprising frequencies of $49, 79, 112, 148, 201, 283,$ and $388$ Hz. While in the SWellEx-96 experiment the position of the source changes with time, for this task we consider the instant when the source depth is 60 m and the distance (or range) between the source and the VLA is 3 km.

The sediment layer is modeled with the following properties. The seabed consists initially of a sediment layer that is 23.5 meters thick, with a density of 1.76 g/cm$^3$, and an attenuation of 0.2 dB/kmHz. The sound speed at the bottom of this layer is assumed to be 1593 m/s. The second layer is mudstone that is 800 meters thick, possessing a density of 2.06 g/cm$^3$, and an attenuation of 0.06 dB/kmHz. The top and bottom sound speeds of this layer are 1881 m/s and 3245 m/s respectively. The description of the geoacoustic model of the SWellEx-96 experiment is complemented by a half-space featuring a density of 2.66 g/cm$^3$, an attenuation of 0.020 dB/kmHz, and a sound speed of 5200 m/s.

Here, we consider two geoacoustic inversion tasks:

*Task 1.* Based on the measurements at the VLA, the objective of this task is to infer the posterior distribution over the water depth as well as the sound speed at the water-sediment interface. For this task, we assume all the quantities above to be known. The unknown parameters $m_1$ (the water depth) and $m_2$ (the sound speed at the water-sediment interface) follow a uniform prior in $[200.5, 236.5]$ m and $[1532, 1592]$ m/s, i.e. $m_1 \sim \mathcal{U}([200.5, 236.5])$ and $m_2 \sim \mathcal{U}([1532, 1592])$, where $\mathcal{U}(\Omega)$ denotes a uniform distribution in the domain $\Omega$.

*Task 2.* In addition to the two parameters considered in *Task 1* (i.e. the water depth $m_1$ and the sound speed at the water-sediment interface $m_2$), we also estimate the posterior distribution over the VLA tilt $m_3$, as well as the thickness of the first (sediment) layer $m_4$. All other quantities provided above are assumed to be known. As in *Task 1*, the unknown parameters follow a uniform prior, i.e. $m_1 \sim \mathcal{U}([200.5, 236.5])$, $m_2 \sim \mathcal{U}([1532, 1592])$, $m_3 \sim \mathcal{U}([-2, 2])$, and $m_4 \sim \mathcal{U}([18.5, 28.5])$.

The received pressure $\mathbf{y}$ on each hydrophone and for each frequency is a function of unknown parameters $\mathbf{m}$ (e.g. water depth, sound speed at the water-sediment interface, etc.) and additive noise $\boldsymbol{\epsilon}$ as follows

$$\mathbf{y} = s(\mathbf{m}, \boldsymbol{\epsilon}) = F(\mathbf{m}) + \boldsymbol{\epsilon}, \qquad \boldsymbol{\epsilon} \sim \mathcal{N}(\mathbf{0}, \boldsymbol{\Sigma}) \qquad (21)$$

where $\boldsymbol{\Sigma}$ is the covariance matrix of data noise. Here, $s(\mathbf{m}, \boldsymbol{\epsilon})$ is a known forward model that, assuming an additive noise model, can be rewritten as $F(\mathbf{m}) + \boldsymbol{\epsilon}$, where $F(\mathbf{m})$ represents the undersea acoustic model Jensen et al. (2011). The SWellEx-96 experiment setup involves a complicated environment and no closed from analytical solution is available for $F(\mathbf{m})$. In this case, $F(\mathbf{m})$ can only be evaluated numerically, and we use the normal-modes program KRAKEN Porter (1992) for this purpose.

Recently, machine learning algorithms have gained attention in the ocean acoustics community (Bianco et al., 2019) (Ying, 2022) for their notable performance and efficiency, especially when compared to traditional methods such as MCMC. In this work, for the first time, we use the concept of invertible networks to estimate posterior distributions in GI. Particularly appealing is that invertible architectures can replace both the forward propagation model as well as the inversion method.

We now discuss the training of the invertible architectures. We use the uniform prior on parameters $\mathbf{m}$ (described in *Task 1* and *Task 2* above) and the forward model in Eq. (21) to construct a synthetic data set for the SWellEx-96 experimental setup using the normal-modes program KRAKEN. For each parameter $\mathbf{m}$, we have $7 \times 21 = 147$ values for the pressure $\mathbf{y}$ received at hydrophones corresponding to the source's 7 different frequencies and the 21 active hydrophones. For inference, we use a test acoustic signal $\mathbf{y}^*$ that corresponds to the actual parameters values from the SWellEx-96 experiment, where the source is 60 m deep

---

[4]see `http://swellex96.ucsd.edu/`

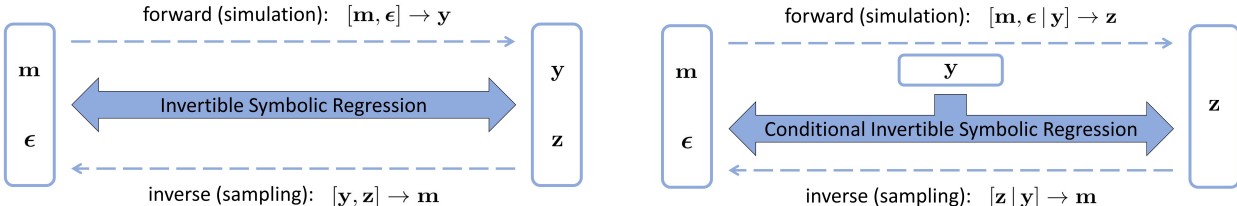

Figure 7: A conceptual figure of ISR (left) and cISR (right) for the geoacoustic inversion task. The posterior distribution of the parameters of interest $\mathbf{m}$ can be obtained by sampling $\mathbf{z}$ (e.g. from a standard Gaussian distribution) for a fixed observation $\mathbf{y}^*$ and running the trained bijective model backwards. To appropriately account for noise in the data, we include random data noise $\boldsymbol{\epsilon}$ as additional model parameters.

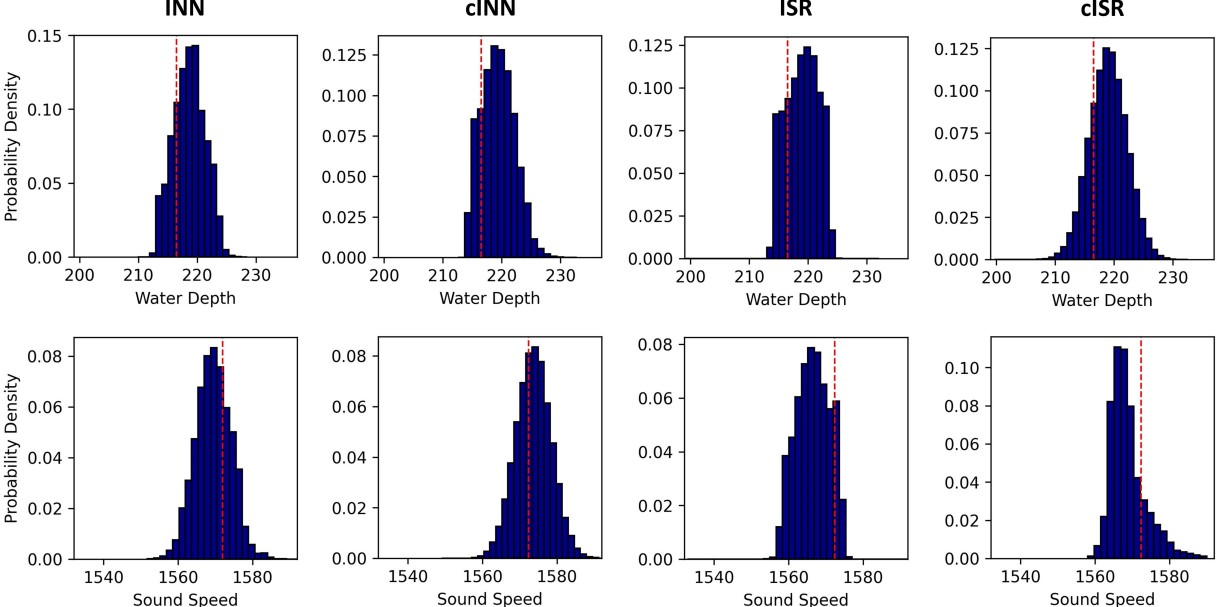

Figure 8: *Task 1.* For a fixed observation $\mathbf{y}^*$, we compare the estimated posteriors $\hat{p}(\mathbf{x} \,|\, \mathbf{y}^*)$ of INN, cINN, and the proposed ISR and cISR methods. Vertical dashed red lines show the ground truth values $\mathbf{x}^*$.

and its distance from the VLA is 3 km (i.e. $m_1^* = 216.5$, $m_2^* = 1572.368$, $m_3^* = 0$, $m_4^* = 23.5$). Also, the signal-to-noise ratio (SNR) is 15 dB.

Inspired by Zhang & Curtis (2021), for the invertible architectures, we include data noise $\boldsymbol{\epsilon}$ as additional model parameters to be learned. In this context, as depicted in the Figure 7, the input of the network is obtained by augmenting the unknown parameters $\mathbf{m}$ with additive noise $\boldsymbol{\epsilon}$, i.e. $\mathbf{x} = [\mathbf{m}, \boldsymbol{\epsilon}]$. There are several ways to use the measurements collected across the 21 hydrophones for training. For instance, one can stack all hydrophones' data and treat them as a single quantity at the network's output. Alternatively, one can treat each hydrophone measurement independently as an individual training example. For the former, the additive noise will be learned separately for each hydrophone pressure $\mathbf{y}$, while for the latter, we essentially learn the effective additive noise over all hydrophones simultaneously. In this experiment, we adopt the latter training approach, which disregards the inter-hydrophone variations, thereby reducing computational overhead.

The pressures received on the hydrophones are considered in the frequency domain, and hence they can be complex numbers. While the invertible architectures can be constructed to address complex numbers, in this case study, we stack the real and imaginary parts of the pressure field at the network's output. That is, the pressure $y = \mathrm{Re}\{y\} + i\,\mathrm{Im}\{y\}$ will be represented as $[\mathrm{Re}\{y\}, \mathrm{Im}\{y\}]$ at the network's output. In short, the 7 pressures (corresponding to the 7 source's frequencies) received at each hydrophone are replaced by 14

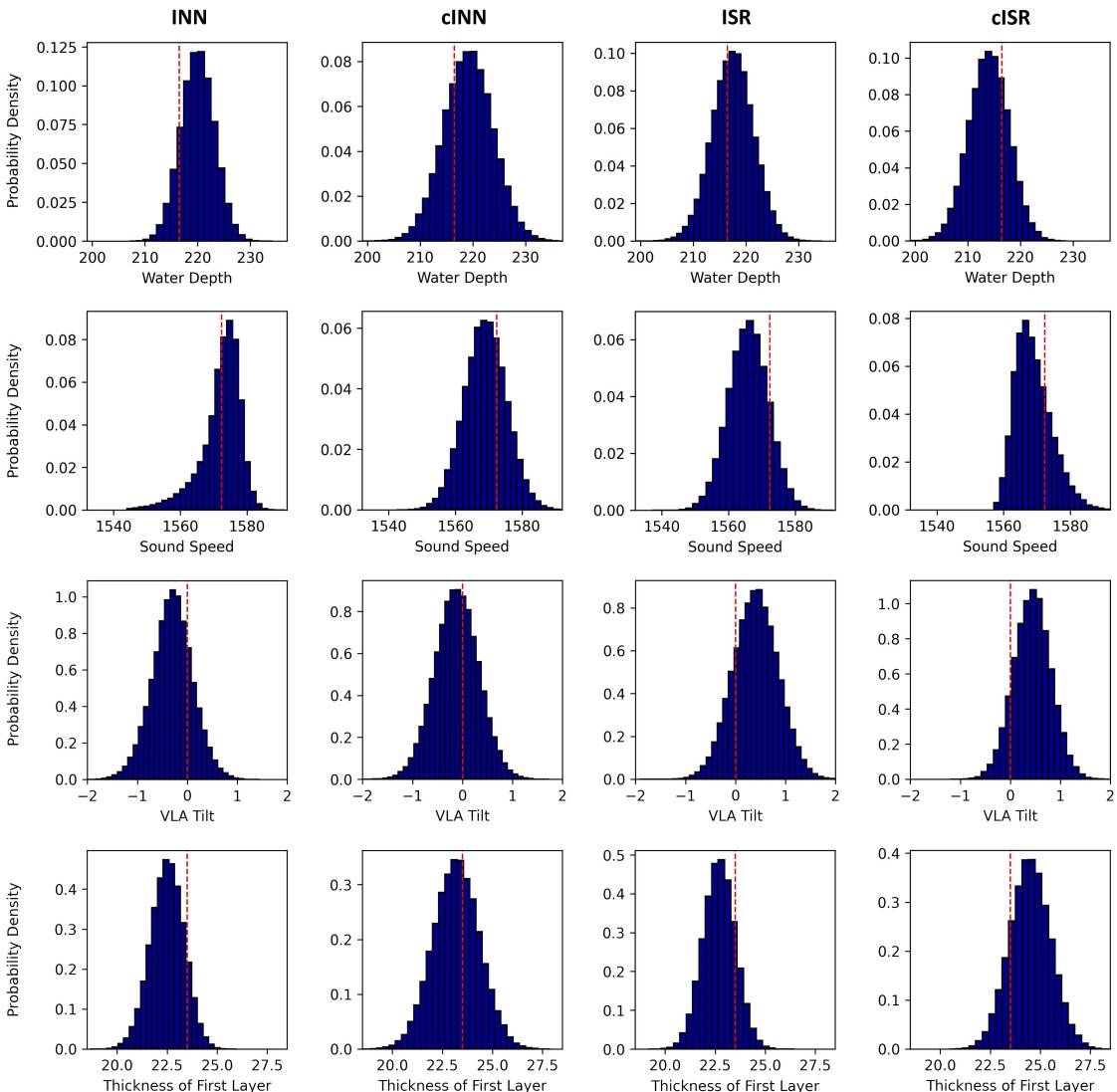

Figure 9: *Task 2.* For a fixed observation $\mathbf{y}^*$, we compare the estimated posteriors $\hat{p}(\mathbf{x}\,|\,\mathbf{y}^*)$ of INN, cINN, and the proposed ISR and cISR methods. Vertical dashed red lines show the ground truth values $\mathbf{x}^*$.

real numbers at the output of the network. Also, the 14 corresponding additive noises are concatenated with the parameters $\mathbf{m}$ at the network's input. Since the dimension of the network's input is $14 + d_{\mathbf{m}}$, the latent variables $\mathbf{z}$ at the network output will be $d_{\mathbf{m}}$-dimensional for ISR and INN, and $(14 + d_{\mathbf{m}})$-dimensional for cINN and cISR. We compare the performance of the proposed ISR and cISR algorithms against INN and cINN in solving GI.

The inferred posterior distributions via INN, cINN, ISR, and cISR, for the GI *Task 1* and *Task 2* are depicted in Fig. 8 and Fig. 9,respectively. The performance of the ISR and cISR architectures is similar to that of the INN and cINN architectures. All methods produce point estimates – Maximum a Posteriori (MAP) estimates – close to the ground truth values, showcasing the efficacy of invertible architectures in addressing cumbersome inversion tasks.

## 5    Conclusion

In this work, we introduce Invertible Symbolic Regression (ISR), a ~~novel~~ technique that identifies the relationships between the inputs and outputs of a given dataset using invertible architectures. This is achieved by bridging and integrating concepts of Invertible Neural Networks (INNs) and Equation Learner (EQL). This integration transforms the affine coupling blocks of INNs into a symbolic framework, resulting in an end-to-end differentiable symbolic inverse architecture that allows for efficient gradient-based learning. The proposed ISR method, equipped with sparsity promoting regularization, has the ability to not only capture complex functional relationships but also yield concise and interpretable invertible expressions. We demonstrate the versatility of ISR as a normalizing flow for density estimation and its applicability in solving inverse problems, particularly in the context of ocean acoustics, where it shows promising results in inferring posterior distributions of underlying parameters. This work is a first attempt toward creating interpretable symbolic invertible maps. While we mainly focused on introducing the ISR architecture and showing its applicability in density estimation tasks and inverse problems, an interesting research direction would be to explore the practicality of ISR in challenging generative modeling tasks (e.g. image or text generation, etc.).

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

# A    Invertible symbolic expressions recovered by ISR for the considered distributions

Table 1: Invertible symbolic expressions recovered by our ISR method for density estimation of several distributions (see Figure 4) in Section 4.1. Here, MoGs denotes "Mixture of Gaussians."

| Example | Expression |
|---------|------------|
| Gaussian | $\mathbf{u} = \mathbf{x}$, i.e. $u_1 = x_1$, $u_2 = x_2$
$s_1(u_2) = 1.16$
$t_1(u_2) = 0$
$v_1 = u_1 \cdot \exp\left(s_1(u_2)\right) + t_1(u_2)$
$v_2 = u_2$
$s_2(v_1) = 1.14$
$t_2(v_1) = -9.39$
$o_1 = v_1$
$o_2 = v_2 \cdot \exp\left(s_2(v_1)\right) + t_2(v_1)$
$\mathbf{z} = \mathbf{o}$, i.e. $z_1 = o_1$, $z_2 = o_2$ |
| Banana | $\mathbf{u} = \mathbf{x}$, i.e. $u_1 = x_1$, $u_2 = x_2$
$s_1(u_2) = 0.52\sin\left(1.86\,u_2\right) + 0.084\sin\left(5.5\sin\left(1.86\,u_2\right) + 2.55\right) + 1.7$
$t_1(u_2) = 0.74 - 0.12\sin\left(3.65\sin\left(2.45\,u_2\right) - 0.82\right)$
$v_1 = u_1 \cdot \exp\left(s_1(u_2)\right) + t_1(u_2)$
$v_2 = u_2$
$s_2(v_1) = 1.72\left(0.025 - 0.47\sin\left(0.33\,v_1\right)\right)\left(0.23\sin\left(0.33\,v_1\right) - 0.29\right) + 2.24$
$t_2(v_1) = -3.74\left(0.022\sin\left(0.62\,v_1\right) + 0.035\sin\left(0.63\,v_1\right) - 0.76\right)\left(0.45\sin\left(0.62\,v_1\right) + 0.7\sin\left(0.63\,v_1\right) + 0.38\right) + 0.027$
$o_1 = v_1$
$o_2 = v_2 \cdot \exp\left(s_2(v_1)\right) + t_2(v_1)$
$\mathbf{z} = \mathbf{o}$, i.e. $z_1 = o_1$, $z_2 = o_2$ |
| Ring | $\mathbf{u} = \mathbf{x}$, i.e. $u_1 = x_1$, $u_2 = x_2$
$s_1(u_2) = -3.14\left(-0.15\,u_2^2 - 0.26\sin\left(1.26\,u_2\right) + 0.094\sin\left(3.25\,u_2\right) - 0.3\right)\left(0.098\,u_2^2 + 0.39\sin\left(1.26\,u_2\right) + 0.014\sin\left(3.25\,u_2\right) + 0.16\right)$
$\qquad + 0.09\sin\left(0.17\,u_2^2 + 0.69\sin\left(1.26\,u_2\right) + 0.76\sin\left(3.25\,u_2\right) + 0.25\right) - 0.30\sin\left(0.94\,u_2^2 + 2.35\sin\left(1.26\,u_2\right) - 1.31\sin\left(3.25\,u_2\right) + 1.57\right) + 0.053$
$t_1(u_2) = -0.012$
$v_1 = u_1 \cdot \exp\left(s_1(u_2)\right) + t_1(u_2)$
$v_2 = u_2$
$s_2(v_1) = 0.037\sin\left(0.22\sin\left(0.22\,v_1\right) + 0.22\sin\left(0.22\,v_1\right) - 0.27\right) + 0.052\sin\left(0.23\sin\left(0.22\,v_1\right) + 0.23\sin\left(0.22\,v_1\right) - 0.39\right) - 0.11$
$t_2(v_1) = 0.13\sin\left(0.13\sin\left(0.37\,v_1\right) + 0.13\sin\left(0.59\,v_1\right) - 1.16\right) + 0.65\sin\left(0.021\,v_1 + 1.34\sin\left(0.37\,v_1\right) + 2.29\sin\left(0.59\,v_1\right) + 1.44\right) + 0.33$
$o_1 = v_1$
$o_2 = v_2 \cdot \exp\left(s_2(v_1)\right) + t_2(v_1)$
$\mathbf{u} = \mathbf{o}$, i.e. $u_1 = o_1$, $u_2 = o_2$
$s_1(u_2) = -3.65\left(-0.47\sin\left(1.38\,u_2\right) - 0.026\sin\left(1.93\,u_2\right) - 0.1\right)\left(-0.014\sin\left(1.38\,u_2\right) - 0.39\sin\left(1.93\,u_2\right) - 0.35\right) - 0.44\sin\left(1.74\sin\left(1.38\,u_2\right) + 1.71\sin\left(1.93\,u_2\right) + 5.55\right)$
$\qquad - 0.46\sin\left(3.31\sin\left(1.38\,u_2\right) + 0.44\sin\left(1.93\,u_2\right) - 0.7\right) + 0.38$
$t_1(u_2) = 0.11\sin\left(0.85\sin\left(1.38\,u_2\right) + 0.86\sin\left(1.38\,u_2\right)\right) + 0.12\sin\left(0.87\sin\left(1.38\,u_2\right) + 0.87\sin\left(1.38\,u_2\right)\right) + 0.053$
$v_1 = u_1 \cdot \exp\left(s_1(u_2)\right) + t_1(u_2)$
$v_2 = u_2$
$s_2(v_1) = 0.25\,v_1^2 - 0.0071\sin\left(1.67\,v_1\right) - 0.1\sin\left(5.29\,v_1\right) - 0.63\sin\left(-1.075\,v_1^2 + 0.55\sin\left(1.67\,v_1\right) + 0.77\sin\left(5.29\,v_1\right)\right)$
$\qquad + 0.46\sin\left(1.89\,v_1^2 - 0.27\sin\left(1.67\,v_1\right) - 1.69\sin\left(5.29\,v_1\right) + 0.9\right) + 1.22$
$t_2(v_1) = 0.62\sin\left(1.56\sin\left(0.43\,v_1\right) + 1.57\sin\left(0.43\,v_1\right) - 1.68\right) + 0.61\sin\left(1.56\sin\left(0.43\,v_1\right) + 1.57\sin\left(0.43\,v_1\right) - 1.68\right) - 0.48$
$o_1 = v_1$
$o_2 = v_2 \cdot \exp\left(s_2(v_1)\right) + t_2(v_1)$
$\mathbf{z} = \mathbf{o}$, i.e. $z_1 = o_1$, $z_2 = o_2$ |
| MoG | $\mathbf{u} = \mathbf{x}$, i.e. $u_1 = x_1$, $u_2 = x_2$
$s_1(u_2) = -0.039\left(-0.032\sin\left(1.44\,u_2\right) - \sin\left(1.48\,u_2\right) + 0.94\right)^2 - 3.03\left(0.18\sin\left(1.44\,u_2\right) + 0.14\sin\left(1.48\,u_2\right) - 0.63\right)\left(0.26\sin\left(1.44\,u_2\right) + 0.28\sin\left(1.48\,u_2\right) - 0.31\right)$
$\qquad + 0.047\sin\left(1.44\,u_2\right) + 0.13\sin\left(1.48\,u_2\right) - 0.029\sin\left(0.15\sin\left(1.44\,u_2\right) + 1.4\sin\left(1.48\,u_2\right) - 3.47\right) - 0.11\sin\left(0.45\sin\left(1.44\,u_2\right) + 1.46\sin\left(1.48\,u_2\right) + 2.65\right) - 0.17$
$t_1(u_2) = 0.052 - 0.12\sin\left(1.13\sin\left(3.14\,u_2\right) + 1.64\right)$
$v_1 = u_1 \cdot \exp\left(s_1(u_2)\right) + t_1(u_2)$
$v_2 = u_2$
$s_2(v_1) = 0.34\left(0.15\sin\left(1.024\,v_1\right) + 0.13\sin\left(2.022\,v_1\right)\right)\sin\left(2.022\,v_1\right) + 0.014\sin^2\left(2.022\,v_1\right) + 0.15\sin\left(0.98\,v_1\sin\left(1.024\,v_1\right) + 1.9\sin\left(2.022\,v_1\right) - 1.41\right)$
$\qquad - 0.22\sin\left(3.38\sin\left(1.024\,v_1\right) + 1.83\sin\left(2.022\,v_1\right) + 1.5\right) + 0.39$
$t_2(v_1) = -0.46\sin\left(0.89\,v_1 + 0.21\sin\left(1.38\,v_1\right) - 1.56\right) + 0.33\sin\left(1.23\,v_1 + 1.69\sin\left(1.38\,v_1\right) - 0.44\sin\left(1.58\,v_1\right) + 1.75\,v_1\right) + 0.094$
$o_1 = v_1$
$o_2 = v_2 \cdot \exp\left(s_2(v_1)\right) + t_2(v_1)$
$\mathbf{u} = \mathbf{o}$, i.e. $u_1 = o_1$, $u_2 = o_2$
$s_1(u_2) = 3.36\left(-0.36\sin\left(3.1\,u_2\right) - 0.29\right)\left(0.26\sin\left(3.1\,u_2\right) + 0.19\right) - 0.45\sin\left(3.88\sin\left(3.1\,u_2\right) - 1.83\right) - 0.38$
$t_1(u_2) = 0$
$v_1 = u_1 \cdot \exp\left(s_1(u_2)\right) + t_1(u_2)$
$v_2 = u_2$
$s_2(v_1) = 0.0035\,v_1^2 - 2.88\left(-0.079\,v_1^2 - 0.33\right)\left(0.042\,v_1^2 + 0.4\right) + 2.6\left(-0.34\sin\left(1.7\,v_1\right) - 0.088\sin\left(1.71\,v_1\right)\right)\left(-0.053\sin\left(1.7\,v_1\right) - 0.38\sin\left(1.71\,v_1\right)\right) + 1.61$
$t_2(v_1) = -0.14$
$o_1 = v_1$
$o_2 = v_2 \cdot \exp\left(s_2(v_1)\right) + t_2(v_1)$
$\mathbf{z} = \mathbf{o}$, i.e. $z_1 = o_1$, $z_2 = o_2$ |

## B    Details of network architectures

In our experiments, we train all models using Adam optimizer with a dynamic learning rate decaying from $10^{-2}$ to $10^{-4}$. In addition, for INN, all (hidden) neurons of the subnetworks are followed by Leaky ReLU activations, while for ISR, each neuron in the hidden layers is followed by an activation function from the following library:

$$\left\{ 1, \mathrm{id}, \bullet^2(\times 4), \sin(2\pi\bullet), \sigma, \bullet_1 \times \bullet_2 \right\}$$

where 1 represents the constant function, "id" is the identity operator, $\sigma$ denotes the sigmoid function, and $\bullet$ denotes placeholder operands, e.g. $\bullet^2$ corresponds to the square operator. Also, $\bullet_1 \times \bullet_2$ denotes the multiplication operator, and each activation function may be duplicated within each layer. We adopt a regularization coefficient of $5 \times 10^{-3}$. The library above is used for illustrative purposes, and indeed, additional arithmetic operators (e.g. $\div$) or mathematical functions (e.g. log, cos, exp, etc.) can be included.

The architecture details are provided below.

*Density estimation via normalizing flow.* In Section 4.1, we train INN and ISR using a batch size of 64. For the "Gaussian" and "Banana" distributions, we adopt 1 affine coupling block with 2 fully connected (hidden) layers per subnetwork. For the "Ring" and "Mixture of Gaussians (MoGs)" distributions, we use 2 invertible blocks with 2 fully connected layers for each subnetwork.

*Inverse Kinematics.* In Section 4.2, we train all models using a batch size of 100. For all models, we adopt 6 reversible blocks with 3 fully connected layers per subnetwork.

*Geoacoustic Inversion.* In Section 4.3, we train all models using a batch size of 200. For all models, we adopt 5 invertible blocks with 4 fully connected layers for each subnetwork.

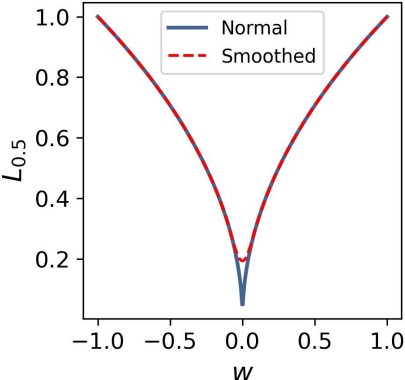

Figure 10: Comparison between the normal $L_{0.5}$ regularization and its smoothed version given by Eq. B.1. For ease of visualization, we set the threshold to $a = 0.1$; however, in our experiments, we adopt a threshold of $a = 0.05$.

**Smoothed $L_{0.5}$ regularization.**    As discussed in Section 2, we use a smoothed $L_{0.5}$ regularization (Wu et al., 2014; Fan et al., 2014; Kim et al., 2020) during training. Figure 10 compares $L_{0.5}$ regularization and its smoothed version. The original (or normal) $L_{0.5}$ regularization creates a gradient singularity as weights approach zero, which can make the training more challenging for gradient based techniques. The smoothed regularization resolves this issue by applying a piecewise function to smooth out the function at small magnitudes, i.e.

$$L_{0.5}(w) = \begin{cases} |w|^{1/2} & |w| \geq a \\ \left( -\frac{w^4}{8a^3} + \frac{3w^2}{4a} + \frac{3a}{8} \right)^{1/2} & |w| < a \end{cases} \tag{B.1}$$

# C   Quantitative Evaluation – Inverse Kinematics

Here, we quantitatively evaluate the quality of the estimated posteriors by the different models considered in the inverse kinematics and the Geoacoustic inversion experiments. To ensure a fair comparison among all methods, we use the same training data and train all models for the same number of epochs, and using identical batches and architectures (as provided in the previous section).

## C.1   Inverse Kinematics

As suggested in (Kruse et al., 2021), we evaluate the correctness of the estimated posteriors using two metrics. First, we use the Maximum Mean Discrepancy (MMD) introduced by Gretton et al. (2012), which computes the *posterior mismatch* between the distribution $\hat{p}(\mathbf{x}\,|\,\mathbf{y}^*)$ produced by a model and a ground truth estimate $p_{\text{gt}}(\mathbf{x}\,|\,\mathbf{y}^*)$, which in this case is obtained via rejection sampling (see Section 4.2), i.e.

$$\text{Err}_{\text{post}} = \text{MMD}\big(\hat{p}(\mathbf{x}\,|\,\mathbf{y}^*), p_{\text{gt}}(\mathbf{x}\,|\,\mathbf{y}^*)\big) \tag{C.1}$$

Second, we measure the *re-simulation error*, which applies the true forward process $f$ in Eq. (20) to the generated samples $\mathbf{x}$ and computes the mean squared distance to the target $\mathbf{y}^*$, i.e.

$$\text{Err}_{\text{resim}} = \mathbb{E}_{\mathbf{x}\sim\hat{p}(\mathbf{x}\,|\,\mathbf{y}^*)}\big[||f(\mathbf{x}) - \mathbf{y}^*||_2^2\big] \tag{C.2}$$

Table 2: Quantitative results for the inverse kinematics benchmark experiment.

| Method | $\text{Err}_{\text{post}}$ | $\text{Err}_{\text{resim}}$ |
|--------|------|------|
| INN | 0.0259 | 0.0163 |
| cINN | 0.0162 | 0.0087 |
| ISR | 0.0286 | 0.0196 |
| cISR | 0.0221 | 0.0134 |

## C.2   Geoacoustic Inversion

To evaluate the estimated posteriors of the unknown parameters (for both *Task 1* and *Task 2* from Section 4.3), we focus on the correctness of Maximum a Posteriori (MAP) estimates (of unknown parameters) $\hat{\mathbf{m}}$ by computing their root-mean-square distance from ground-truth values $\mathbf{m}^*$ over test set observations $\mathbf{y}^*$, i.e.

$$\text{Err}_{\text{MAP}} = \sqrt{\mathbb{E}_{\mathbf{y}^*}\big[||\hat{\mathbf{m}} - \mathbf{m}^*||_2^2\big]}. \tag{C.3}$$

Table 3: Quantitative results for the geoacoustic inversion experiment.

| Method | *Task 1 (2-D)* $\text{Err}_{\text{MAP}}$ | *Task 2 (4-D)* $\text{Err}_{\text{MAP}}$ |
|--------|------|------|
| INN | 3.822 | 3.981 |
| cINN | 3.503 | 3.743 |
| ISR | 5.101 | 5.574 |
| cISR | 4.597 | 4.993 |

# D    Illustrative Example

In this section, we consider an interesting toy inverse problem to illustrate the challenges and opportunities of the INN and ISR methods. We consider the following forward model

$$y = x^2 + \epsilon \tag{D.1}$$

where the input $x$ and output $y$ are scalar quantities. Here, we also consider the additive noise $\epsilon \sim \mathcal{N}(0, 0.1)$, and we assume a standard normal prior on the input $x$. A closed-form inverse solution exists for this toy inverse problem and, as shown in Figure 11, the posterior $p(x\,|\,y)$ is bimodal.

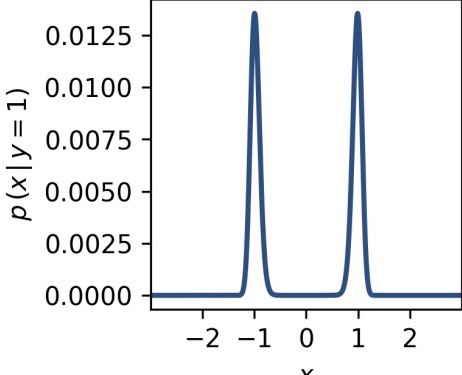

Figure 11: True posterior $p(x\,|\,y^*)$, conditioned on $y^* = 1$, for the forward model described in Eq. (D.1). Indeed, for $y^* = 1$, two inverse solutions are possible, i.e. $x = \pm 1$, which explains the bimodal shape.

As discussed in Section 3.2, to use coupling-based invertible architectures with one-dimensional (or scalar) input, we pad the input with an extra zero and utilize a loss term that prevents the encoding of information in the extra dimension. The expressive power of ISR (and cISR) depends on the library of arithmetic operators and mathematical functions being used, which is a design choice. Here, we adopt the library given in Appendix B. Indeed, one can use a different library when necessary or when domain knowledge is available. The estimated posteriors using INN, cINN, ISR, and cISR, are depicted in Figure 12. The invertible symbolic expressions generated by ISR and cISR are reported in Table 4.

Interestingly, as illustrated in Figure 12, initial observations indicate that all methods provide a solution to the specified inverse problem. Additionally, it is noted that the INN, cINN, and cISR methods capture the bimodal shape of the distribution, whereas the ISR method identifies only one mode. Notably, the posterior predicted by cISR appears broader than those predicted by INN and cINN, which align closely with the true posterior. This phenomenon can be attributed to the inherent trade-off between accuracy and interpretability in SR methods. In this instance, the application of sparsity-promoting smoothed $L_{0.5}$ regularization in training cISR yields a relatively simple and interpretable model, as demonstrated in Table 4. However, this simplicity comes at the expense of accuracy, as evidenced by the broader posterior distribution observed with cISR.

Another interesting observation is that the solution recovered by ISR primarily produces a posterior with only one mode. Examination of Table 4 reveals that ISR accurately recovered the analytical expression for the forward model, specifically $y = x^2$. However, given that ISR employs a fixed invertible architecture characterized by affine coupling blocks, the exact analytical expression for the inverse function may not always be attainable. The inverse map approximation that ISR generates is straightforward and interpretable, yet it does not precisely match the expected expression (i.e., $x = \pm\sqrt{y}$). This discrepancy likely contributes to the unimodal shape observed in the predicted posterior distribution.

Indeed, the choice of library used for implementation can significantly influence the results. Additionally, the $L_{0.5}$ regularization coefficient plays a critical role in determining the sparsity of the resulting symbolic

solution. It is important to note that even with the same library and regularization settings, different potential symbolic expressions (or solutions to the inverse problem) may be derived from those shown, and these expressions could exhibit either unimodal or bimodal distributions based on the produced approximations.

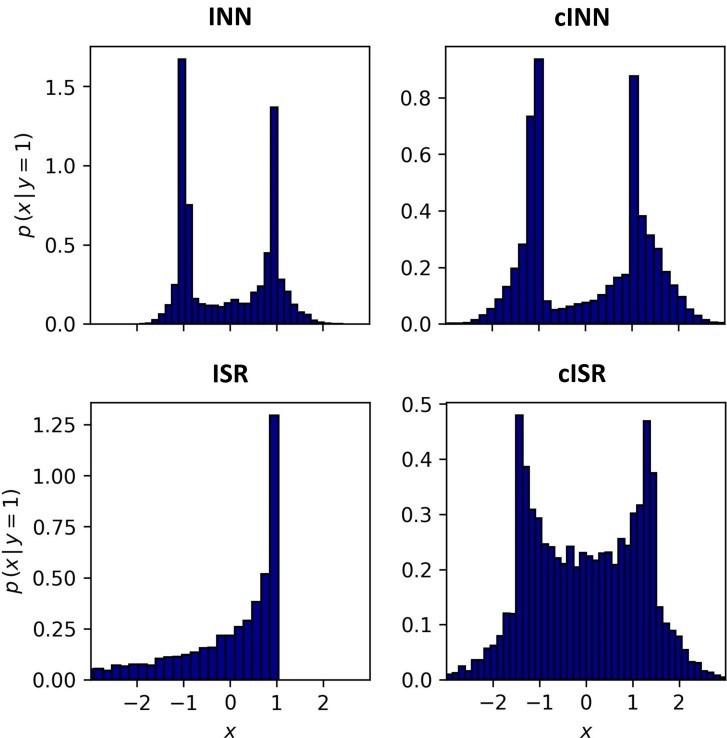

Figure 12: For a fixed observation $y^* = 1$, we compare the predicted posteriors $\hat{p}(x \,|\, y^*)$ of INN, cINN, and the proposed ISR and cISR methods.

Table 4: Invertible symbolic expressions recovered by our ISR and cISR methods for the toy example above.

| Method | Expression |
|--------|------------|
| ISR | $\mathbf{u} = [0, \mathbf{x}]$, i.e. $u_1 = 0$, $u_2 = x$ |
| | $s_1(u_2) = 0.0018u_2^2 + 0.012u_2 + 0.107942 \sin{(0.51u_2)} - 0.039$ |
| | $t_1(u_2) = u_2^2$ |
| | $v_1 = u_1 \cdot \exp{(s_1(u_2))} + t_1(u_2)$ |
| | $v_2 = u_2$ |
| | $s_2(v_1) = 0.0024v_1^2 + 0.011v_1 - 0.13 \sin{(1.69v_1)} - 0.17$ |
| | $t_2(v_1) = -0.0052v_1^2 - 0.0092v_1 - 0.101 \sin{(0.92v_1)} + 0.039 \sin{(1.38v_1)} - 0.036$ |
| | $o_1 = v_1$ |
| | $o_2 = v_2 \cdot \exp{(s_2(v_1))} + t_2(v_1)$ |
| | $[y, z] = \mathbf{o}$, i.e. $y = o_1$, $z = o_2$ |
| cISR | $\mathbf{u} = [0, \mathbf{x}]$, i.e. $u_1 = 0$, $u_2 = x$ |
| | $s_1(u_2, y) = 5.11$ |
| | $t_1(u_2, y) = 0.65u_2 - 0.013y - 0.01 \left(-u_2 - 0.61y\right)^2 + 0.025 \left(-u_2 + 0.35y\right)^2 + 0.0087$ |
| | $v_1 = u_1 \cdot \exp{(s_1(u_2, y))} + t_1(u_2, y)$ |
| | $v_2 = u_2$ |
| | $s_2(v_1, y) = 0.3v_1^2 + 0.00071 \left(-0.19v_1 - y\right)^2 + 4.48$ |
| | $t_2(v_1, y) = -134.037v_1 - 0.39y - 4.56 \left(-v_1 - 0.37y\right)^2 - 3.79 \left(-v_1 - 0.37y\right)^2 - 6.38 \left(-v_1 - 0.37y\right)^2$ |
| | $\qquad + 8.64 \left(v_1 - 0.36y\right)^2 + 4.97 \left(v_1 - 0.35y\right)^2 + 4.703 \left(v_1 - 0.35y\right)^2 - 1.89 \left(v_1 + 0.37y\right)^2 + 0.15$ |
| | $o_1 = v_1$ |
| | $o_2 = v_2 \cdot \exp{(s_2(v_1, y))} + t_2(v_1, y)$ |
| | $\mathbf{z} = \mathbf{o}$, i.e. $z_1 = o_1$, $z_2 = o_2$ |

