# OpenReview forum: "ISR: Invertible Symbolic Regression"
_TMLR — Rejected by TMLR_

### Review · Reviewer_mL16 · 2024-06-16

**Summary Of Contributions:**

Authors propose the "invertible symbolic regression" framework to establish the mapping between the observables $\mathbf{y}$ and latent variables $\mathbf{z}$ in the concatenated form $[\mathbf{y},\mathbf{z}]$ with the underlying variables $\mathbf{x}$. The method is largely based on the normalized flow and prototypes proposed in Ardizzone et al., 2019 ICML work.

**Audience:**

Yes

**Claims And Evidence:**

No

**Requested Changes:**

(1) Authors open their argument by mentioning that _the objective of SR is to find an analytical expression $f$ that best maps inputs to outputs_ and that _in ISR, we aim to learn a bijective symbolic function_ $f: \mathbb{R}^{d_{\mathbf{x}}}\to \mathbb{R}^{d_{\mathbf{y}}} \times \mathbb{R}^{d_{\mathbf{z}}}$ _from the space of functions ..._.

I do not get why there is any "search" of optimal function forms or components ongoing in the proposed methodology that echoes this proclaimed motivation.

(2) Besides point (1), considering all preliminary functions such as arithmetic operations, trigonometric functions, exponential and logarithmic functions, it is unclear that all these function forms are included in the EQL network.

(3) My biggest question is: I do not see what necessitates or motivates to use this invertible mapping framework. Does it outperform other non-invertible functional forms in any way in the context of solving inverse problems? Or does it bear any unique advantages in any special use cases? The current presentation of the numerical result does not seem to be conclusive.

(4) What are the specifics of the _equation learner networks_ exactly implemented in this work?

(5) In the application section 4.3, can authors report similar metrics as in [Ardizzone et al. ICML 2019]?

(6) Inverse acoustic scattering problems have an established thread of work dedicated to it. See section 3.3 in [Solving Inverse Problems with Deep Learning, Lexing Ying, 2021] for example and the references therein. Comparison with other established inverse problem solvers to benchmark with the normalized flow-based methods seem to be constructive in making clear why this approach should be favoured.

**Strengths And Weaknesses:**

General comments with detailed feedback listed in the following sections:

(1) The motivation of resorting to the invertible structure for solving inverse problems is unclear in this submitted manuscript.

(2) The novel contribution distinguished from [Ardizzone et al., Analyzing inverse problems with invertible neural networks, ICLR, 2019] and two other papers by Ardizzone et al. is unclear.

(3) The experimental results do not seem to comprehensively reflect the advantages of the proposed methods.

(4) Some argument paragraphs in the main text seem convoluted while unable to make clear points.

---

> ### Author Response · Authors · 2024-07-16
> **Response to Reviewer mL16**
>
> We thank the reviewer for providing valuable comments which helped us improve the quality of the manuscript.
>
> > (1) The motivation of resorting to the invertible structure for solving inverse problems is unclear in this submitted manuscript.
>
> Resorting to invertible maps or architectures for solving inverse problems has been widely adopted in the literature (Ardizonne et al. 2019; Kruse et al. 2021, Zhang and Curtis, 2021, etc.). That said, we agree with the reviewer that the motivation in the beginning of the manuscript is not clear, and hence we have revised the paper accordingly.
>
> > (2) The novel contribution distinguished from [Ardizzone et al., Analyzing inverse problems with invertible neural networks, ICLR, 2019] and two other papers by Ardizzone et al. is unclear.
>
> Our method combines (Ardizonne et al., 2019) with Symbolic Regression concepts realized by EQL networks, resulting in a new ISR framework. \
> This work is the first of its kind. There is no existing approach that can perform Symbolic Regression for inverse problems.\
> Our method indeed relies on existing components, but the recipe to combine invertible architectures with EQL networks to create an invertible symbolic framework is completely new. \
> We would like to note that the general SR problem by itself is already a very challenging task that is extensively investigated by the ML community and that the development of an Invertible Symbolic Regression (ISR) method is highly nontrivial.
>
> > (3) The experimental results do not seem to comprehensively reflect the advantages of the proposed methods.
>
> The advantages of the ISR method are first shown through the normalizing flow example. Both INN and ISR produce an invertible model (the flow), but ISR provides a symbolic expression, which is clearly an advantage.\
> In addition, the other experiments demonstrate that ISR also scales to higher dimensional problems where INN is useful, proving that ISR can easily replace INN while also providing some interpretability to the solution.\
> Indeed, when the dimension of the problem increases, we lose the interpretability as the model becomes more complex. However, complex analytical models could still be more interpretable than inscrutable black box neural networks.\
> Why would someone prefer INN over ISR if both generate same results?
>
> > (4) Some argument paragraphs in the main text seem convoluted while unable to make clear points.
>
> We have reviewed the paper and made revisions to enhance readability and flow.
>
> __Requested Changes:__
> > (1) Authors open their argument by mentioning that _the objective of SR is to find an analytical expression $f$ that best maps inputs to outputs and that in ISR, we aim to learn a bijective symbolic function $f: \mathbb{R}^{d_x} \rightarrow \mathbb{R}^{d_y} \times \mathbb{R}^{d_z}$ from the space of functions ..._.\
> I do not get why there is any "search" of optimal function forms or components ongoing in the proposed methodology that echoes this proclaimed motivation.
>
> The general Symbolic Regression problem is to search the space of functions to find the optimal analytical expression given data. However, many methods in the literature, especially the parametric ones, assume some general symbolic structure, and only parameters are optimized, as a way to increase learning efficiency and reduce computational time. Traditionally, SR methods rely on Genetic Programming (GP) to search for the optimal symbolic expression. However, GP-based methods usually come at a higher computational cost.
>
> Here in the manuscript, we first introduce the general objective of ISR, which is essential similar to that of the general SR problem with the additional constraint that the resulting model has to be invertible. As a way to illustrate the proposed method, we used EQL networks whose parametrized nature enhances computational efficiency. Indeed, Genetic Algorithms or other existing SR frameworks could also be integrated within the invertible architecture, at a possibly higher computational cost.
>
> We have added a paragraph in the paper to address this point and update the claims/argument.

---

> > ### Author Response · Authors · 2024-07-16
> > **Continuation of Response to Reviewer mL16**
> >
> > > (2) Besides point (1), considering all preliminary functions such as arithmetic operations, trigonometric functions, exponential and logarithmic functions, it is unclear that all these function forms are included in the EQL network.
> >
> > The EQL network used in this paper, by design, allows for all arithmetic operators except division (i.e. $+$, $-$, $\times$). Similar to the multiplication operator, the division operator can be integrated within the EQL network, but this could lead to numerical instabilities, which is the case for most SR methods. In addition, for illustrative purposes, we used the library of mathematical functions provided in Appendix B. That said, the user can indeed increase the expressive power of the method by including other functions (e.g. log, cos, etc.) within the library when necessary or when domain knowledge is available, especially in real-world applications. We have added a note in Appendix B to address this point.
> >
> > > (3) My biggest question is: I do not see what necessitates or motivates to use this invertible mapping framework. Does it outperform other non-invertible functional forms in any way in the context of solving inverse problems? Or does it bear any unique advantages in any special use cases? The current presentation of the numerical result does not seem to be conclusive.
> >
> > To the best of our knowledge, this is the first work that uses Symbolic Regression to establish invertible analytical models from data. Our approach, naturally combines symbolic regression concepts (EQL networks) with invertible maps and architectures. This leads to an invertible symbolic framework, providing interpretability to normalizing flows.
> >
> > The main objective of the paper is not to solve inverse problems or to compare with non-invertible functional forms; here the goal is to introduce this ISR approach, which can provide a new feature to symbolic regression models by ensuring their invertibility. This is especially useful for solving inverse problems in the fields of controls or physics. Future work aims at applying the ISR method to a system identification problem in a controls setting.
> >
> > > (4) What are the specifics of the equation learner networks exactly implemented in this work?
> >
> > The specifics of the EQL networks are provided in Appendix B. Additionally, we have included the regularization coefficient value, which was inadvertently not specified in the original manuscript.
> >
> > > (5) In the application section 4.3, can authors report similar metrics as in [Ardizzone et al. ICML 2019]?
> >
> > We updated Appendix C accordingly.
> >
> > > (6) Inverse acoustic scattering problems have an established thread of work dedicated to it. See section 3.3 in [Solving Inverse Problems with Deep Learning, Lexing Ying, 2021] for example and the references therein. Comparison with other established inverse problem solvers to benchmark with the normalized flow-based methods seem to be constructive in making clear why this approach should be favoured.
> >
> > We would like to thank the reviewer for pointing us to this line of works. We have added a paragraph with the following references in the manuscript:
> > * Solving Inverse Problems with Deep Learning, Lexing Ying, 2021.
> > * Solving inverse wave scattering with deep learning, Yuwei Fan and Lexing Ying, 2019.
> > * SwitchNet: a neural network model for forward and inverse scattering problems, Yuehaw Khoo and Lexing Ying, 2019.

---

### Review · Reviewer_5pNo · 2024-07-01

**Summary Of Contributions:**

This article proposes a framework for learning bijective transformations which are interpretable in terms of mathematical equations. Based on existing works on normalizing flows (condition/unconditional), the article proposes to construct invertible neural networks based on EQL (equation learner). This allows one to learn interpretable normalizing flows. The main contribution of this article is to make this idea work in the context of density estimation, and on practical inverse problems by solving a conditional density estimation problem. A technical challenge is to impose a sparse regularization on EQL to learn simple models. Numerical results show a good performance with respect to existing normalizing flows.

This article seems to be a first attempt to make normalizing flows interpretable (I am not aware of related works on this). If this is true, I tends to accept the article. If not, a citation (and comparison) should be added.

**Audience:**

Yes

**Claims And Evidence:**

No

**Requested Changes:**

Questions:
- Discuss more in detail how the 0.5 sparse regularization works, how sensitive it is the result to this regularization.
- I am wondering if the framework proposed in Figure 2 is the same in the context of normalizing flows. If we replace the ISR in Figure 2 by an invertible neural work, the rest of the framework (such as objective functions) seems to be the same. Could you clarify the novelty of the framework? I find that it is a simple replacement of a network architecture, thus this is not really a new framework.
- The ISR equation learnt in the Gaussian case of Figure 4 is very interesting. But it seems to be a very special case. Could you give more such interpretable results on other cases? To make the point convincing, I’d suggest to have at least 2 examples where the obtained ISR makes sense.
- A detailed question on section 4.2, why you use a Gaussian distribution on x to represent some angular variable which should be periodic? It seems to me to use a uniform distribution makes more sense.
- Also, could you give more details on the INN network in this case, to see if there is big difference compared to the ISR ?
- Make the variable x specific in Section 4.3.

**Strengths And Weaknesses:**

Strength:

- The numerical results are promising to illustrate the capacity of such networks to approximate simple and complex probability densities.

- One technical difficulty is about how to impose sparse constraints on EQL. I think this should be discussed in more detail in the article.

- The article is well written, easy to understand.

Weakness:

- Novelty of the proposed ISR framework seems not so significant.

- The learnt EQL does not seem to be so interpretable.

---

> ### Author Response · Authors · 2024-07-16
> **Response to Reviewer 5pNo**
>
> We thank the reviewer for recognizing the novelty and contribution presented in the paper, and for providing valuable suggestions and comments to improve the exposition of the paper!
>
> > This article seems to be a first attempt to make normalizing flows interpretable (I am not aware of related works on this). If this is true, I tends to accept the article. If not, a citation (and comparison) should be added.
>
> To the best of our knowledge, this is the first work to try combining symbolic regression concepts with invertible maps and architectures, to create an invertible symbolic framework, providing interpretability to normalizing flows.
>
> __Weaknesses:__
>
> > * Novelty of the proposed ISR framework seems not so significant.
>
> We agree that the individual components we use are not new, but the recipe to naturally combine invertible architectures with EQL networks to create symbolic invertible framework is the first of its kind. We would like to note that the symbolic regression problem by itself is already a challenging task that is extensively investigated by the machine learning community. The proposed Invertible SR (ISR) method is both novel and highly relevant for real-world applications. More generally, the presented invertible SR framework provides a pathway to solve inverse problems based on existing SR methods. Thus, as future SR approaches become available, our framework makes it possible to directly employ them for inverse problems.
>
> > * The learnt EQL does not seem to be so interpretable.
>
> Most of Symbolic Regression methods have a trade-off between accuracy and interpretability, hence the more accurate the model is, the less interpretable it might be. The definition of interpretability can have different meanings for different experts or stakeholders, and hence having a not-so-simple analytical expression would still be more interpretable than inscrutable black box neural networks. In addition, having an analytical expression provides access to gradients in closed-form as well, which can be of importance in many applications.
>
> __Requested Changes:__
>
> > * Discuss more in detail how the 0.5 sparse regularization works, how sensitive it is the result to this regularization.
>
> We have added a detailed discussion about the 0.5 sparse regularization in the Appendix.
>
> > * I am wondering if the framework proposed in Figure 2 is the same in the context of normalizing flows. If we replace the ISR in Figure 2 by an invertible neural work, the rest of the framework (such as objective functions) seems to be the same. Could you clarify the novelty of the framework? I find that it is a simple replacement of a network architecture, thus this is not really a new framework.
>
> First, we are not claiming novelty of the framework, but we are the first to attempt finding invertible analytical expressions through a clever combination of established algorithmic building blocks. Indeed, if EQL networks are not used, then the method would reduce to the original standard invertible neural network architecture. However, as we mentioned above, by simply integrating EQL networks into the invertible architecture, we are able to produce an invertible symbolic framework that can provide interpretability to inverse problems solution and normalizing flows. We updated/reduced the claims accordingly in the manuscript.
>
> Finally, it is also worth mentioning that our approach aligns well with TMLR's scope, which emphasizes technical correctness over subjective significance.
>
> > * The ISR equation learnt in the Gaussian case of Figure 4 is very interesting. But it seems to be a very special case. Could you give more such interpretable results on other cases? To make the point convincing, I’d suggest to have at least 2 examples where the obtained ISR makes sense.
>
> We have added an interesting example in the Appendix. Please check the updated manuscript.
>
> > * A detailed question on section 4.2, why you use a Gaussian distribution on x to represent some angular variable which should be periodic? It seems to me to use a uniform distribution makes more sense.
>
> We agree with the reviewer that a uniform distribution could make more sense in this context, however, we preferred to stick to the original benchmark and experimental setup used in the literature including the works in (Ardizzone et al. 2019; Kruse et al. 2021).
>
> > * Also, could you give more details on the INN network in this case, to see if there is big difference compared to the ISR ?
>
> Details of the network architectures (including INN) are provided in Appendix B. The main difference lies within the use of a standard NNs versus symbolic EQL networks equipped with sparsity promoting regularization.
>
> > * Make the variable x specific in Section 4.3.
>
> Thank you for pointing this out. We have added this specification in Section 4.3 (i.e. x = [m, $\epsilon$]).

---

### Review · Reviewer_7ZW9 · 2024-07-02

**Summary Of Contributions:**

The paper introduces Invertible Symbolic Regression (ISR), which merges concepts from Invertible Neural Networks (INNs) and Equation Learner (EQL) to tackle interpretability in inverse problems. ISR aims to derive interpretable mathematical relationships between system parameters and observable quantities using symbolic architectures, facilitating explicit computation of posterior probabilities in complex systems.

**Audience:**

Yes

**Claims And Evidence:**

No

**Requested Changes:**

It could be useful to 1) address the lack of the logarithm operation, and 2) demonstrate that the proposed method can recover a given analytical expression based on its sampled data.

**Strengths And Weaknesses:**

Strengths:
1. The proposed ISR combines symbolic regression with invertible neural network architectures, allowing for the derivation of interpretable mathematical expressions that describe the relationships between parameters and observables.
2.  ISR generates models that are both accurate and interpretable, distinguishing it from traditional black-box neural networks. This is achieved by optimizing for sparse symbolic expressions, aiding in understanding the underlying physical laws governing the system.
3. The paper demonstrates ISR's effectiveness across domains like inverse kinematics and geoacoustic inversion. These examples showcase ISR's ability to handle diverse inverse problems while maintaining interpretability.

Weaknesses:
1. The expressive power of the method could be limited due to the lack of necessary operators. For example, the architecture's omission of logarithmic operations limits its expressive power for problems where logarithms are essential. This could restrict ISR's applicability to certain real-world scenarios.
2. The absence of weight regularization in ISR could lead to overly complex expressions that are functionally equivalent but unnecessarily convoluted. This might compromise model interpretability and the ability to discern the simplest solution.
3. While ISR shows promise in density estimation tasks, its advantages in interpretability are not extensively validated beyond a simple Gaussian distribution. Evaluating ISR's ability to recover analytical expressions from sampled data could clarify its uniqueness, identifiability, and generalization capabilities.

---

> ### Author Response · Authors · 2024-07-16
> **Response to Reviewer 7ZW9**
>
> We thank the reviewer for the positive feedback and helpful comments.
>
> __Weaknesses:__
>
> > 1. The expressive power of the method could be limited due to the lack of necessary operators. For example, the architecture's omission of logarithmic operations limits its expressive power for problems where logarithms are essential. This could restrict ISR's applicability to certain real-world scenarios.
>
> The expressive power of ISR depends on the library of operators being used, which is a design choice. In this paper, for illustrative purposes, we used the library mentioned in Appendix B. However, the user can indeed add other operators (e.g. log, etc.) when necessary or when domain knowledge is available, as typically the case in real-world scenarios. We have added a note in Appendix B to address this point.
>
> > 2. The absence of weight regularization in ISR could lead to overly complex expressions that are functionally equivalent but unnecessarily convoluted. This might compromise model interpretability and the ability to discern the simplest solution.
>
> The weight regularization is a key factor in ISR, as it is in most if not all existing SR methods, and indeed its absence will lead to more complex solutions. In practice, most SR methods have some sort of trade-off between accuracy and complexity in order to produce the simplest solution possible that still provides a high accuracy.
>
> > 3. While ISR shows promise in density estimation tasks, its advantages in interpretability are not extensively validated beyond a simple Gaussian distribution. Evaluating ISR's ability to recover analytical expressions from sampled data could clarify its uniqueness, identifiability, and generalization capabilities.
>
> Recovering analytical expressions from sampled data is the objective of the standard SR formulation in general, and is usually a challenging task especially when the data is noisy  $-$ a common challenge in real-world scenarios.
>
> In the proposed ISR method, we are adding the constraint that the resulting expression has to be invertible, hoping that the resulting invertible symbolic expression provides accurate (or even exact) mappings in both the forward and the inverse directions (which is even more challenging).
>
> Although ISR might recover the exact analytical expression (which is typically defined as the forward direction), the predefined/fixed invertible architecture (which is based on the affine coupling blocks) used in ISR may or may not recover the exact analytical expression for the inverse direction. The approximation of the inverse map that is recovered will influence the shape of the posterior distribution. We have added an interesting example in the Appendix to illustrate this point.
>
> __Requested Changes:__
>
> > It could be useful to 1) address the lack of the logarithm operation, and 2) demonstrate that the proposed method can recover a given analytical expression based on its sampled data.
>
> We have revised and updated the manuscript accordingly to address the reviewer's concerns.

---

### Decision · Action_Editor_Ca8i · 2024-08-11

**Recommendation:** Reject

**Comment:**

While the revisions made by the authors in response to the reviews are a step in the right direction, they do not address some key issues with the paper. The paper needs an exploration of the accuracy-interpretability trade-off for the ISR solutions and a much more convincing demonstration of their claimed interpretability.

**Audience:**

I am not sure there is an audience for the paper in its current form. The primary factor that distinguishes this work from Ardizzone et al., 2019 is the claimed interpretability of the discovered solutions, but as mentioned above this is not sufficiently demonstrated in the paper. Moreover, there is hardly any insight provided into the method. Most notably, there is no exploration of the effect of the sparsity regularization coefficient that presumably controls the trade-off the accuracy and interpretability of the solutions, and without such an exploration the paper cannot be considered complete and is of limited interest. The paper would also be strengthened by discussing the limitations of adopting the coupling-based invertible approach and explaining why the authors believe it is well suited for producing interpretable solutions when combined with ELNs.

**Claims And Evidence:**

The authors introduce a method, called Invertible Symbolic Regression (ISR), for solving inverse problems using "symbolic" invertible models, building on Ardizzone et al., 2019. The main contribution is replacing the feedforward neural nets (FFNNs) in the coupling blocks of invertible models with Equation Learner Networks (ELNs), which are essentially FFNNs with nonlinearities replaced with primitive functions such as multiplication, squaring, sine, etc. The models are trained using maximum likelihood with sparsity regularization to encourage sparser (and thus simpler) ELN solutions. The authors show that the proposed method performs on par or slightly worse than the method of Ardizzone et al., 2019 for comparable model sizes. The major weakness of the paper is that while the approach is motivated by promising interpretability of the learned mappings, as the reviewers pointed out, this is not convincingly demonstrated. The Gaussian example in Section 4.1 is nice, but it is a very simple special case, and the solutions for other tasks presented in the paper do not seem interpretable in any meaningful sense of the word.